# Human nucleolar protein SURF6/RRP14 participates in early steps of pre-rRNA processing

Anastasiia Moraleva[1]*, Alexander Deryabin[1], Maria Kordyukova[2], Mikhail Polzikov[1], Kseniya Shishova[2], Kira Dobrochaeva[1], Yury Rubtsov[1], Maria Rubtsova[3], Olga Dontsova[1,3,4], Olga Zatsepina[1†]

1 Shemyakin-Ovchinnikov Institute of Bioorganic Chemistry RAS, Moscow, Russian Federation, 2 Federal Center of Brain Research and Neurotechnologies of the Federal Medical Biological Agency, Moscow, Russian Federation, 3 Department of Chemistry, Lomonosov Moscow State University, Moscow, Russian Federation, 4 SkolTech, Moscow, Russian Federation

☯ These authors contributed equally to this work.
† Deceased.
* a_moraleva@mail.ru

**Data Availability Statement:** All relevant data are within the paper and its Supporting information files.

## Abstract

The biogenesis of ribosomes requires tightly controlled transcription and processing of pre-rRNA which comprises ribosomal RNAs forming the core of large and small ribosomal sub-units. Early steps of the pre-rRNA processing and assembly of the ribosomal subunits require a large set of proteins that perform folding and nucleolytic cleavage of pre-rRNAs in the nucleoli. Structure and functions of proteins involved in the pre-rRNA processing have been extensively studied in the budding yeast *S. cerevisiae*. Functional characterization of their human homologues is complicated by the complexity of mammalian ribosomes and increased number of protein factors involved in the ribosomal biogenesis. Homologues of human nucleolar protein SURF6 from yeast and mouse, Rrp14 and Surf6, respectively, had been shown to be involved in the early steps of pre-rRNA processing. Rrp14 works as RNA chaperone in complex with proteins Ssf1 and Rrp15. Human SURF6 knockdown and over-expression were used to clarify a role of SURF6 in the early steps of pre-rRNA processing in human cell lines HeLa and HTC116. By analyzing the abundance of the rRNA precursors in cells with decreased level or overexpression of SURF6, we demonstrated that human SURF6 is involved in the maturation of rRNAs from both small and large ribosomal subunits. Changes in the SURF6 level caused by knockdown or overexpression of the protein do not result in the death of HeLa cells in contrast to murine embryonic fibroblasts, but significantly alter the distribution of cells among the phases of the cell cycle. SURF6 knockdown in both p53 sufficient and p53 deficient HCT116 human cancer cells results in elongation of G0/G1 and shortening of G2/M phase. This surprising result suggests p53 independence of SURF6 effects on the cell cycle and possible multiple functions of SURF6. Our data point to the shift from pathway 1 to pathway 2 of the rRNA biogenesis caused by the SURF6 knock-down and its likely association with p53 pathway.

**Funding:** This work was funded by RFBR (Russian Foundation for Basic Research) grant No. 20-04-00796 "Analysis of the protein-nucleic acid composition of ribosomal subunit assembly intermediates in genetically modified human cells". The funder had no role in study design, data collection and analysis, decision to publish, or preparation of the manuscript.

**Competing interests:** The authors have declared that no competing interests exist.

## Introduction

Human SURF6/RRP14 is a nucleolar protein from an evolutionarily conserved family. SURF6 homologues have been found in eukaryotes from yeasts to humans [1]. In mammals, SURF6 gene is located in the Surfeit locus, and its protein product was found in all tissues [2]. SURF6 level in murine cells depends on the stage of cell cycle [3]. Interestingly, in murine and human T cells from spleen and peripheral blood, SURF6 is normally expressed at low level, but its amount progressively increases following the stimulation of lymphocytes by mitogens [4, 5]. Accordingly, the amount of SURF6 is significantly higher in peripheral blood lymphocytes of patients with lymphoproliferative diseases in contrast to healthy donors [5].

A putative function of human SURF6 was proposed based on the structural similarity between human SURF6 and its yeast homologue Rrp14 [1]. Rrp14 is involved in different stages of ribosome biogenesis from 35S pre-rRNA synthesis to large subunit assembly [6, 7]. Cryo-EM studies revealed that Rrp14 forms a complex with two other proteins, Ssf1 and Rrp15 (homologues of the human PPAN and RRP15, respectively), which plays a role of chaperone that facilitates folding of domains I and VI of 28S rRNA. Domains I and VI form two sides of the polypeptide exit tunnel in the mature ribosome. Before cleavage of ITS2 (Internal Transcribed Spacer 2), the triple complex Rrp14-Ssf1-Rrp15 is replaced by the protein Rpl31 [8, 9]. Depletion of Rrp14 caused by genetic knockout in yeast results in almost complete arrest of 20S pre-rRNA (the precursor of 18S rRNA) and 27S pre-rRNA processing (the precursor of 25S rRNA) and leads to cell death. At the same time, it was shown that SURF6 knockout in yeast causes accumulation of aberrant A2–C2 fragments. Perhaps, Rrp14 could be responsible for inhibition of premature cleavage of the C2 site in ITS2 in direct manner or/and by binding/affecting other factors but no other steps in pre-rRNA processing [7].

Previously, we and others showed that knockdown of SURF6 in mouse embryonic fibroblasts leads to cell death [10], whereas its overexpression promotes the proliferation of cells accompanied by accumulation of the 32S pre-rRNA [11]. These data suggest that human SURF6 should be also involved in processing of pre-rRNA and assembly of the ribosomal subunits. In contrast to primates, nucleolus in mice, despite close evolutionary relationship of rodents and primates, lacks several intrinsic elements. It was found that there is a difference in cleavage at some pre-rRNA sites in human and mouse [12]. In primate's nucleoli, there are species-specific proteins [13] and lncRNAs [14], which are involved in the organization of the nucleolus structure. Also, some nucleolar proteins, in particular NMP1 (protein partner of SURF6) have been found only in primates [14–16]. Therefore, the exact role of SURF6 in human ribosome biogenesis remains unclear and deserves further studies.

Here, we characterized the functional role of the human SURF6 in the ribosome biogenesis. The effects of SURF6 siRNA knockdown and overexpression in human HeLa cell line were investigated. Our results demonstrate that in comparison to primary human cells neither overexpression nor knockdown is lethal to HeLa cells. Unexpectedly, this relatively mild phenotype is associated with dramatic changes in the profile of the pre-rRNA intermediates as well as significant redistribution of cells among the stages of the cell cycle. We report that the effect of SURF6 on cell cycle and cell survival is independent of p53 as follows from the study of p53 sufficient and deficient HCT116 cells. We report specific stages of human pre-rRNA processing that require SURF6 and demonstrate the difference between human and yeast SURF6 in respect of pre-rRNA processing.

## Materials and methods

### Plasmid constructs

E. coli strain XL 1-Blue (Stratagene, USA) was propagated in LB media supplemented with antibiotics and used for routine DNA cloning according to standard protocol [17]. The CDS (RefSeq NM_006753.6) of human SURF6 (1092 b.p.) was PCR amplified using Ecyclo DNA polymerase (Evrogen, Russia), total cDNA from HeLa cell lines and specific primers: 5'-cgg aattctgatggcctctctactcgccaaggac-3' (direct), 5'-gcgggatcctcagacca ggcctgcgct' (reverse). Amplified fragment digested by *BamH*I and *Eco*RI was ligated into pcDNA3.1 vector («Clontech», USA) linearized with the same restriction enzymes. Resulting pcDNA3.1-hSurf6 was isolated, purified with Plasmid Midi Kit (Qiagen, USA) and sequenced. Empty pcDNA3.1 vector was used as negative control in SURF6 overexpression experiments.

### Cell culture and transfection

HeLa cells were obtained from Russian bank of cell lines (Institute of Cytology RAS, Saint-Petersburg, Russia), HCT116 p53+ and HCT116/p53- [18] cells were kindly provided by Olga Dontsova lab (Department of Chemistry, Lomonosov Moscow state University). Cells were grown in DMEM (PanEco, Russia) supplemented with 10% fetal bovine serum (HyClone, USA), 2 mM L-glutamine, and penicillin/streptomycin (250 units/ml, all PanEco, Russia) at 37˚C and 5% $CO_2$. Sub-confluent HeLa were transfected with plasmid constructs using Lipofectamine 2000 (Thermo, USA) according to vendor's protocol. Transfected cells were analyzed 48 hours post transfection. For siRNA-mediated knockdown of SURF6 cells were transfected with siRNA SURF6 RNA duplexes (Life technologies, USA; anti-SURF6 siRNA id S13655, neg. control id am4635) using Lipofectamine 2000 (Life technologies, USA) and corresponding protocol. At day 3 post transfection, cells were re-seeded and transfected with siRNA again. Cells were harvested 24–72 hours post the second transfection and analyzed.

### Northern blotting

Total RNA from HeLa cells was isolated using Trizol (Thermo, USA) according to standard protocol. RNA quality and concentration were assessed by spectrophotometer Genesis 10 Bio (Termo Electron Co., USA) at 260/280 nm wavelengths. RNA was separated by gel electrophoresis in 1.2% agarose gel containing 1,8M formaldehyde in 20mM MOPS/ 4mM NaAc/ 0,5mM EDTA (pH 7.0) as a buffer. 1 **μ**g of total RNA was loaded per lane and run at 2V/cm for 4 hours. RNA was transferred to BrightStar®-Plus Positively Charged Nylon Membrane (Life technologies, USA) according to alkali transfer protocol for 2 hours. 1M NaCl/10mM NaOH was used as a transfer buffer. Membrane was rinsed in 2x SSC (0,3M NaCl, 30 mM Sodium Citrate pH 7.0) and baked at 80˚C for 15 min in the oven. Hybridization with the biotin-labeled probes (10 ng/ml final concentration) was performed in ULTRAhyb®-Oligo buffer (Life technologies, USA) according to manufacturer's instruction at 42˚C for 12–16 hours. The probe's sequences were as following: 5'ITS2-biotin-GGGGCGATTGATCGGCAAG CGACGCTC, 5'ITS1-biotin-GGCCTCGCCCTCCGGGCTCCG, 7SK-biotin-GACGCA CATGGAGCGGTGAGGGAG. After hybridization the membrane was washed twice with 2xSSC, 0,5% SDS for 30 min at 42˚C. Bands were visualized using Chemiluminescent Nucleic Acid Detection Module (Pierce, USA) at room temperature according to manufacturer's instructions.

The X-ray film was scanned, and the resulting image was processed using the Matlab software package. Each line was processed separately, the intensity of band signal was integrated over the width of each lane. Signal level was corrected using the intensity of the signal-free

areas of the image. The resulting curves were fitted with a set of Gaussian peaks with given width constraints. The resulting line values corresponded to the integral under the Gaussian peak minus the base line value.

## Analysis of rRNA precursors ratio

RAMP (Ratio Analysis of Multiple Precursors) profiles were generated according to [19], the density of rRNA precursor bands was determined as described previously. The ratio of precursors was calculated for each total RNA sample (pre-rRNA 41S/45-47S, 12S/32S etc), each ratio was obtained from the analysis of single lane following the hybridization with one probe. $\log_2$ values of precursors ratios in the control samples (untreated HeLa and HCT116 cells) were subtracted from $\log_2$ values of corresponding ratios in samples from cells with SURF6 knockdown or overexpression. Resulting normalized $\log_2$ values for SURF6 knockdown and overexpression were plotted as histograms and combined two different ratios in one RAMP profile. The first one is the ratio of corresponding pre-rRNA to the primary rRNA transcript, and the second is the ratio between pre-rRNAs which form substrate-product pairs, for example, 18SE/21S and 12S/32S.

## SDS-PAGE and Western blotting

$1 \times 10^6$ HeLa or HCT116 p53+/p53- cells were lysed on ice in 200 **μ**L of buffer containing 50 mM Tris-HCl (pH 7,5), 150 mM NaCl, 10% glycerol, 1% Triton X-100, 1 mM PMSF, 1 mM DTT). Protein concentration in lysates was determined by Bradford reaction according to standard protocol. Lysate samples were diluted with 5x Laemmli buffer and subjected to SDS-PAGE (25 ug per lane) in 12% gel [20] prior to transfer to nitrocellulose membrane (BioRad, USA). Membrane was blocked in 5% non-fat dry milk in TBST (20 mM Tris-HCl, pH 7,6, 150 mM NaCl, 0,1% Tween 20) for 1 hour at room temperature and incubated with anti-SURF6 [21] or anti-actin (Sigma, A4700) or α-tubulin (Sigma, T6199) or anti-p53 (Abcam, ab131442) primary antibodies in TBST 5% non-fat dry milk for 1 hour at room temperature. After three 5-minute washes in TBST the membrane was incubated with secondary anti-mouse or anti-rabbit (in the case of p53) HRP-conjugated IgG (Sigma, 1:20000 in TBST/ 5% non-fat dry milk) for 1 hour at room temperature. Then the membrane was washed three times in TBST, treated with ECL reagents (Amersham Pharmacia Biotech. Inc., Great Britain) according to vendor's instruction, prior to band visualization using X-ray film or ChemiDoc imager (BioRad).

## Immunocytochemistry

HeLa cells were grown on cover slides. Then cells were fixed either in acetone or paraformaldehyde prior to staining procedure.

Cells were fixed in absolute acetone at -20˚C for 10 min, which corresponds to the optimal conditions for detecting SURF6 at the cellular level. Cells were stained with a mixture of two antibodies: rabbit anti-SURF6 (1:100 dilution) and mouse anti-B23/nucleophosmin (1:200 dilution; Sigma-Aldrich, B0556) for 1 hour at room temperature [22]. Then cells were washed in PBS for 3x5 min and stained for 1h at room temperature with a mixture of AlexaFluor 488 goat anti-rabbit IgG (HCL) antibodies (Molecular Probes Inc., USA, cat.A-11034; for detection of SURF6) and Alexa Fluor® 568 goat anti -mouse IgG (H+L; Molecular probes Inc., cat. A11004; for detection of B23/nucleophosmin).

Cells were fixed with 2% paraformaldehyde in PBS buffer (140 mM NaCl, 2.7 mM KCl, 1.5 mM KH2PO4, and 8.1 mM $Na_2HPO_4$, pH 7.2) for 20 min at room temperature. Then cells were treated with 0.5% Triton X-100 in PBS for 10 min at room temperature, and

stained with a mixture of antibodies: mouse antibodies to SURF6 [21] (diluted 1:100) and rabbit antibodies to Fibrillarin (diluted 1:200; Abcam, ab5821) for 1h at room temperature. Then washed for 3x5 min in PBS and stained with a mixture of antibodies Alexa Fluor® 488 goat anti-mouse IgG (H+L; Molecular probes Inc., cat. A11029; for detection of SURF6) and Alexa Fluor® 568 goat anti-rabbi IgG (H+L; Molecular probes Inc., cat. A11011; for detection of Fibrillarin).

In both cases, after staining with secondary antibodies, cells were washed with PBS for 3x5 min, stained with DNA-binding dye DAPI (1 mg/ml in PBS, 10 min) and fixed in Vectashield (Vector Laboratories, USA). The preparations were analyzed using a DuoScan-Meta LSM510 confocal laser scanning microscope (Carl Zeiss, Germany) equipped with a Plan-Apochromat 63 x1.40 (numerical aperture) oil Ph3 objective.

## FISH

Following biotin-labeled oligonucleotide probes used for FISH: ITS1 5' biotin-GGCCTCG CCCTCCGGGCTCCG, ITS2 5'biotin-GGGGCGATTGATCGGCAAGCGACGCTC, 28S 5'biotin-CCTCTTCGGGGGACGCGCGCGTGGCCCCGA, 18S 5'biotin- ATCGGCCC GAGGTTACTCTAGAGTCACCAAA. Cells were fixed with 4% paraformaldehyde for 30 min, washed three times with PBS for 5 min and treated with 0.5% Triton-X100/PBS for 10 min at $4^o$ C. After 3x washing in PBS for 5 min each, followed by 5 min wash in 2xSSC (0,3M NaCl, 30 mM Sodium Citrate pH 7.0, the probes diluted in deionized formamide (Sigma) and pre-heated to 70°C for 10 min were added to hybridization mix (10% dextran sulfate (Sigma), 2xSSC, final concentration of probe 8 ng/ml). Samples were incubated for 16h at 42°C in moisturized chamber. After hybridization the samples were washed first with 50% formamide (Pancreac, Spain) in 2xSSC 3 times 10 min each at 42°C, then once with 2×SSC for 10 at 42°C, and once with 2xSSC at the room temperature. Biotinylated probes were stained with rhodamine-conjugated streptavidin (Roche, USA) for 1h at room temperature. After three 10 min washes in 4×SSC (0.6M NaCl, 60 mM $Na_3C_6H_5O_7$) supplemented with 0.05% Tween-20 and single wash in PBS for 10 min cells were incubated with anti-SURF-6 antibodies. Samples were washed in PBS prior to addition of Alexa488- or AMCA-conjugated anti-mouse IgG (Molecular probes and Jackson Labs, respectively), and addition of DAPI (see figure legend). Slides were washed with PBS, embedded into Mowiol, the images were acquired using fluorescent Axiovert 200 microscope equipped with mercury lamp and digital camera.

Confocal overlapping images of green and red fluorescent signals was obtained by Ar-Kr-Ne laser excitation at 488 and 543 nm. Each channel was scanned separately to avoid spillover of signals caused by spectral overlap.

## Analysis (quantification) of co-localization of pre-rRNAs, rRNAs and SURF6

To quantify the extent of co-localization of RNAs and SURF6 1286128-pixel digital images of nucleoli were taken, and for each pair of probes at least 10 images were analyzed. The algorithm by [23] was utilized in ImageJ package (NIH, Bethesda, MD) equipped with plug-in JACoP [24, 25]. Intensity of pixels in one channel was juxtaposed to the intensity in other channel of double colored image. For channel intensity comparison, a scatterplots were created and the linear equation of the channels was calculated using linear regression. According to the scatterplots, the Pearson correlation coefficients and Manders co-localization

coefficients were calculated [26].

$$r = \frac{\left(\sum i(Ri - Rcp)(Gi - Gcp)\right)}{\sqrt{\left(\sum (R - Rcp)^2 \sum (G - Gcp)^2\right)}}$$ (1)

M1 was defined as the ratio of net pixel intensity in green channel with non-zero red intensity in red channel to net intensity in green channel, M2 was calculated in the same way. Accordingly, M1 defines the part of signal in green channel which comes from red channel.

$$M1 = \frac{\sum iRicoloc}{\sum iRi}$$ (2)

$$M2 = \frac{\sum iGicoloc}{\sum iGi}$$ (3)

where $R_{i\ coloc} = 0$, if $G_i = 0$, $R_{i\ coloc} = R_i$ if $G_i > 0$  $G_{i\ coloc} = 0$, if $R_i = 0$, $G_{i\ coloc} = G_i$ if $R_i > 0$.

Values range from 0 to 1, representing the fraction of the intensity in a channel, measured in pixels where the intensity is higher than the threshold in another color channel. These coefficients measure the correlation indicating the degree of joint localization. The thresholds that the Costes method sets means that the correlation below the thresholds is zero. In this case, the compatibility is measured independently of the proportionality of the signal. In the Costes method, the threshold value required for background identification is determined automatically based on an analysis that determines the range of pixel values for which a positive Pearson covariance is obtained. Pearson's coefficient is measured for all pixels in the image, and then again for pixels with a lower intensity value of both channels on the regression line. This process is repeated until the pixel values are reached for which the Pearson coefficient drops to zero and below. The values of the intensities of red and green on the regression line at this point are then used as threshold values to identify the background level. Only those pixels for which the red and green intensity values are higher than the corresponding threshold values are pixels with jointly localized probes. The Menders coefficients are then calculated, which determine the fraction of fluorescence in the region of interest. The method also includes creating a scattering diagram of the R or G channel relative to the product of the difference between the intensities of the pixels R and G between its corresponding mean values (Ri-Rav) (Gi-Gav) Then the results of the correlation analysis of the intensity are presented in the form of a pair of graphs, where the x-axis reflects the channel covariance, and y is the distribution of the channel intensity. Distributions that are shifted to the right indicate positively correlated patterns, where the coloration values of the two pixels change synchronously. Those that are symmetrical about the 0 axis indicate independent coloration, those shifted to the left are negatively correlated.

## MTT assay

HeLa or HCT116 p53+/p53- cells were seeded in 96-well plates in DMEM/10% FBS three days after the second transfection with siRNA for 4 hours. The MTT reagent was added according to the manufacturer's protocol (Thermo, USA) to the final concentration of 0,5 mg/mL and cells were incubated 4 hours at 37˚ C, 5% CO2. Absorbance was measured at 570 nm using Thermo Scientific Multiskan EX ELISA reader (Thermo Fisher Scientific, USA). Experiments were made in triplicates.

### Flow cytometry

Cells were fixed with 70% ethanol for 15 min at 4°C and stained with 50 mg/ml propidium iodide (Sigma-Aldrich) in PBS containing 0.5 mg/ml bovine pancreatic RNAase A (Sigma-Aldrich) for 1 hour at 37° C. Above $1 \times 10^4$ cells were analyzed in each sample using a flow cytometer Epics"Elite" (Beckman Coulter Inc., USA) equipped with an argon laser Cyonics (Uniphase, USA) and Multigraph software (Coulter, USA). Experiments were repeated four times. Then the data were analyzed using Kaluza Analysis Software (Beckman Coulter Inc., USA).

### Statistical analysis

Results are expressed as mean ± standard error of the mean (SEM). Control and experimental values were compared with the Student t-test assuming unequal variances. Differences are considered significant at a p value less than 0.05. Microsoft Excel 2007 software (Microsoft, Redmond, WA, USA) was used for statistical analysis of data.

## Results

### SURF-6 co-localizes with ITS1, ITS2 and 18S in nucleoli of HeLa cells

In our previous study, we demonstrated that murine Surf6 is involved in ribosome biogenesis, which was not shown for human SURF6. Therefore, we decided to study the co-localization of SURF6 with ITS1, ITS2, 18S and 28S parts of pre-rRNA in nucleoli of fixed/permeabilized HeLa cells using FISH with respective specific probes (S1N Fig, scheme of 47S pre-rRNA and location of probes) and immunostaining of SURF6 (S1 Fig). The goal of this experiment was to estimate at what stage (early or late) of pre-rRNA processing SURF6 might participate. Despite an imperfection of this approach, it is clear from confocal images that signal from SURF6 overlaps with the signal from ITS2 FISH (S1D–S1F Fig). At the same time, SURF6 also significantly co-localizes with the areas of ITS1 FISH signal (about 70% overlap, S1A–S1C Fig). Overlay of SURF6 immunostaining with FISH of 18S and 28S was less prominent (S1G–S1L Fig). Quantitation of overlaps from FISH and SURF6 immunostaining is presented in S1M Fig. The data suggest that SURF6 could potentially form complexes with LSU (large sub-unit) precursors that contain ITS2, and possibly with less mature 41S precursor of both SSU (small subunit) and LSU (includes both ITS1 and ITS2). In addition, the co-localization of SURF6 with 18S and 28S was less prominent suggesting SURF6 participation in the stages of the ribosome biogenesis preceding the early processing of internal transcribed spacers. Unfortunately, the co-localization fails to reveal the exact stages of rRNA processing and biogenesis of ribosomal subunits which require participation of SURF6. Therefore, we decided to set up functional experiments based on analysis of phenotypic changes induced by knockdown and overexpression of SURF6 in HeLa cells.

### SURF6 knockdown and overexpression do not significantly affect cell morphology

SURF6 knockdown was induced by transfection of HeLa cells with siRNA specific to human SURF6 mRNA. Changes in SURF6 expression were assessed by immunoblotting of lysates obtained from SURF6 siRNA-transfected and control scramble siRNA-transfected cells according to established protocol [27]. As expected, SURF6 level dramatically decreased in SURF6 siRNA-transfected cells, but not in control cells. We observed at least 90% knockdown efficiency according to analysis of protein level (Fig 1A and 1B). In contrast to mouse embryonic fibroblasts, where Surf6 siRNA knockdown caused significant changes in cell morphology

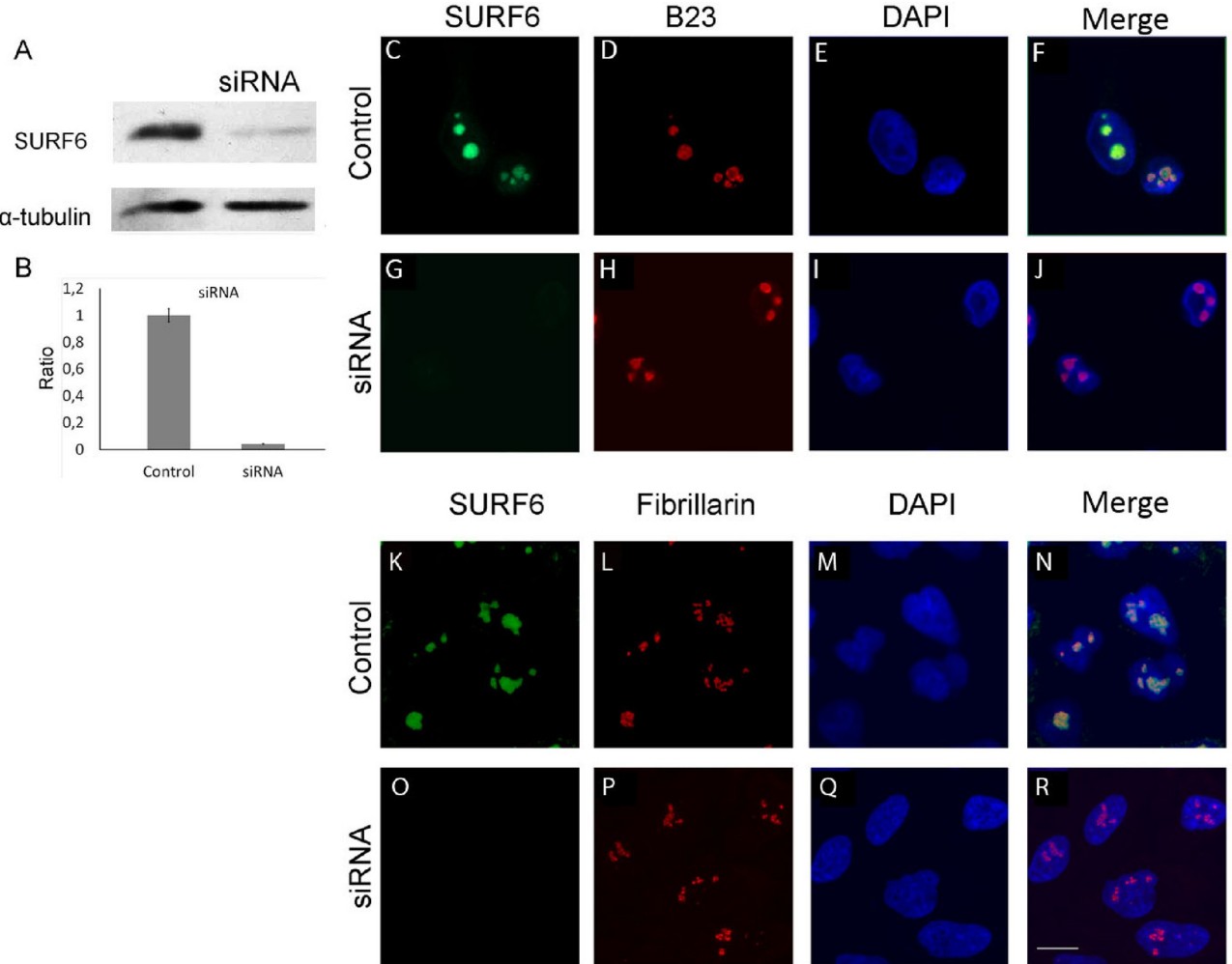

**Fig 1. SURF6 knockdown does not disturb morphology of HeLa cells.** Western blot analysis (A, B) of SURF6 in protein extracts of HeLa cells with si-mediated knockdown of SURF6. Anti-tubulin staining was used for normalization. Confocal microscopy of fixed/permeabilized HeLa cells stained with anti-SURF6 (C, G, K, O), anti-B23 (D, H), anti-fibrillarin (L, P) and DAPI (E, I, M, Q) prior to imaging. (F, J, N, R) merged images.

and eventual cell death [10], SURF6 knockdown in HeLa had no cytotoxic effect according to MTT test neither disturbed cell morphology (Figs 1C–1R and 5H). For overexpression, pcDNA3.1-hSURF6 was introduced into HeLa cells, and changes in SURF6 level in lysates from transiently transfected cells were examined 48 hours post transfection by Western blotting and immunostaining with anti-SURF6 antibodies. Lysate obtained from HeLa cells transfected in parallel with empty vector was used as negative control. As a result, the amount of SURF6 (expected single 46 kDa band) was greatly increased in lysates from HeLa/pcDNA3.1--SURF6 (Fig 2A and 2B). Fluorescence intensity of nucleoli in the HeLa/pcDNA3.1-SURF6 stained with anti-SURF6 was markedly higher than that in control cells (Fig 2C, 2G, 2K and 2O). According to our estimation, SURF6 level was approximately 5 times higher in HeLa/pcDNA3.1-SURF6 than in control (Fig 2B). Microscopic images clearly demonstrate that neither overexpression nor knockdown do not affect predominantly nucleolar localization of SURF6 (Figs 1C–1R and 2C–2R). It should be noted that despite significant difference in SURF6 level between HeLa with overexpression and knockdown of SURF6, we failed to find

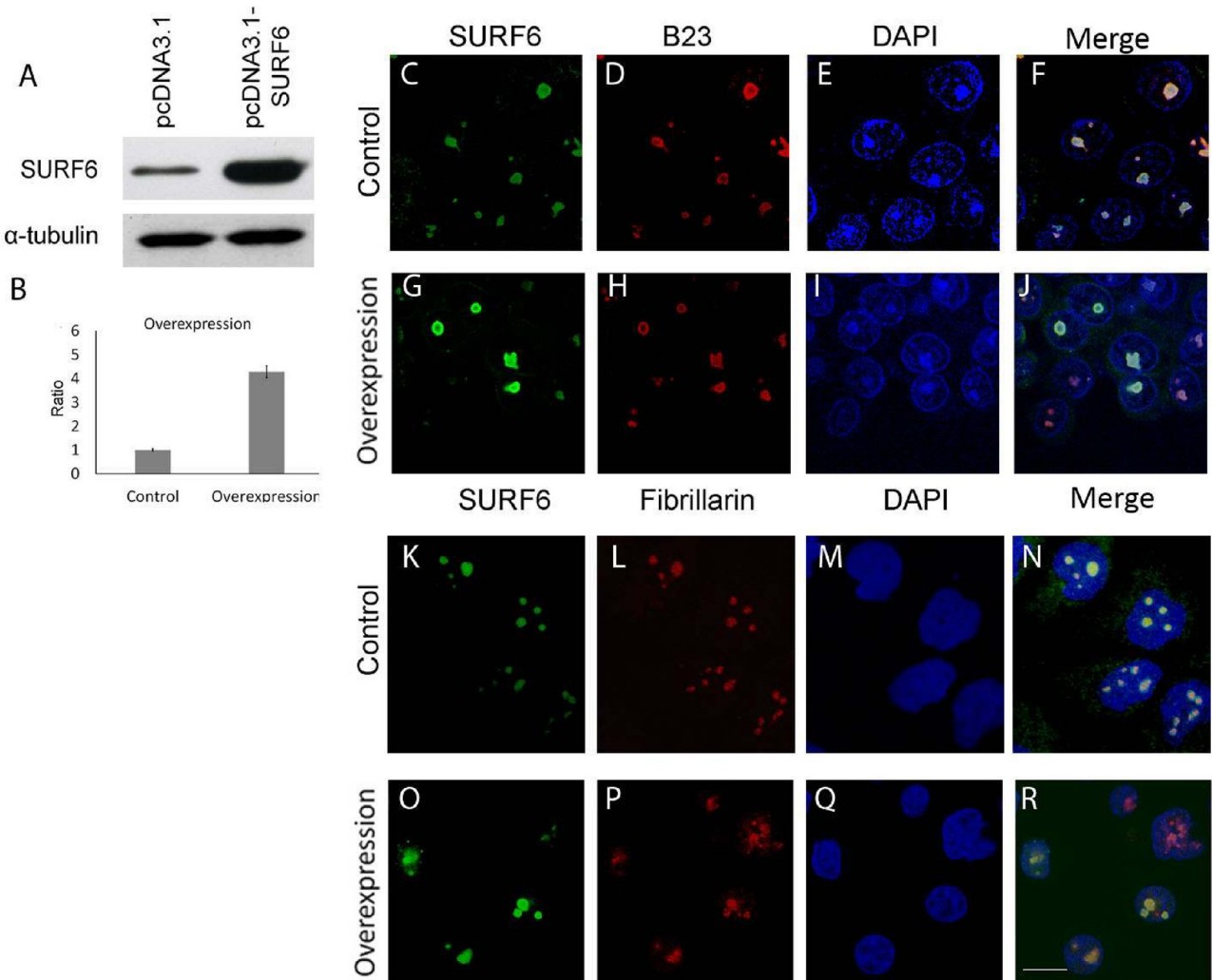

**Fig 2. SURF6 overexpression does not disturb morphology of HeLa cells [31].** Western blot analysis (A, B) of SURF6 in protein extracts of HeLa cells with overexpression of SURF6. Anti-tubulin staining was used for normalization. Confocal microscopy of fixed/permeabilized HeLa cells stained with anti-SURF6 (C, G, K, O), anti-B23 (D, H), anti-fibrillarin (L, P) and DAPI (E, I, M, Q) prior to imaging. (F, J, N, R) merged images.

detectable changes in size, shape, and pattern of DNA staining in nuclei (Figs 1C–1R and 2C–2R). To examine potential changes in fine architecture of nucleolus, we co-stained nucleoli of control HeLa, SURF6 knockdown and overexpression HeLa with anti-SURF6 and anti-fibrillarin (nucleolar fibrillar component marker) or anti-SURF6 and anti-B23 (nucleolar granular component marker). Confocal images demonstrate that fine patterning of fibrillarin and B23 are not affected by both SURF6 knockdown (Fig 1C–1R) and overexpression (Fig 2C–2R). It is interesting that fibrillarin signal is more punctate than that of SURF6 suggesting SURF6 could potentially function outside fibrillar centers [16]. These results suppose that overexpression of human SURF6 in HeLa does not disturb cell morphology in contrast to Surf6 overexpression in mouse embryonic cells [11]. Formally, it is not possible to exclude that there are faint changes in nucleoli of HeLa cells following overexpression or knockdown of SURF6, but their visualization would require high-resolution EM studies.

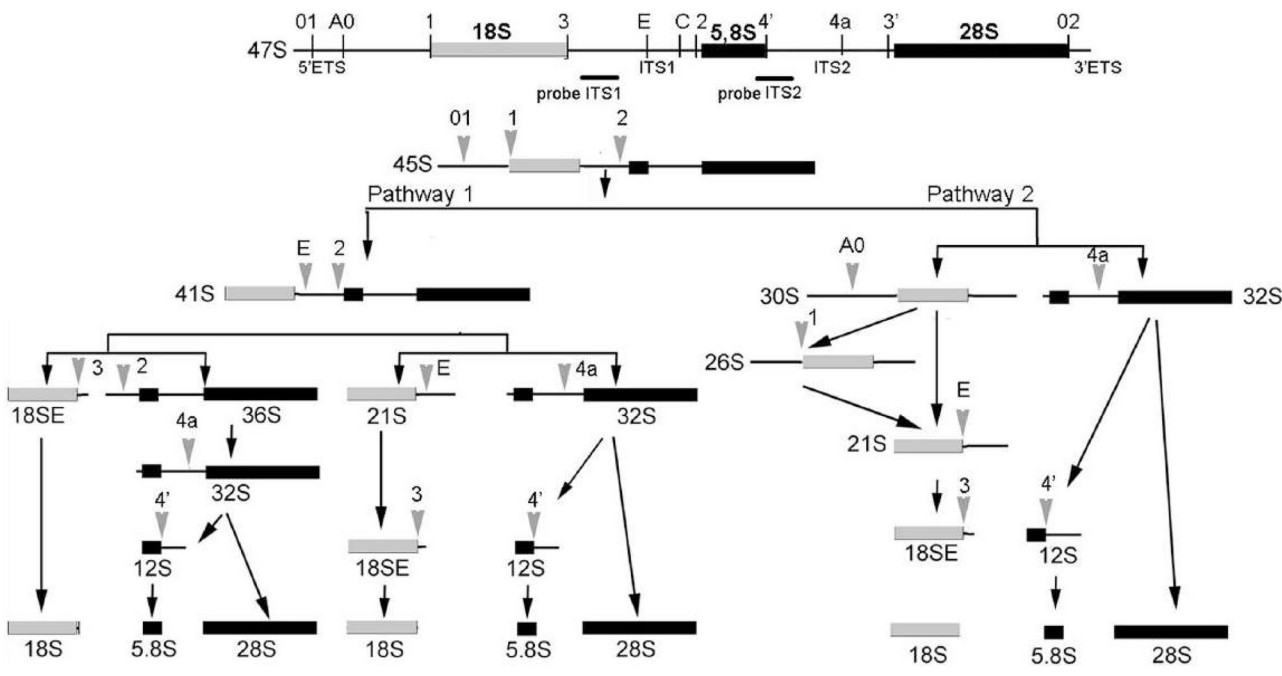

**Fig 3. Pre-rRNA processing pathways in mammalian cells [30].**

## Both SURF6 knockdown and overexpression change pre-rRNA profile

As it was mentioned before, yeast and murine homologues of Surf6 are involved in the maturation of ribosomal subunits. To compare the roles of yeast Rrp14 discussed in [6–8, 28] and human SURF6 in early stages of the ribosomal biogenesis (schematically shown in Fig 3), Northern blot analysis of different pre-rRNAs from HeLa/pcDNA3.1-SURF6, HeLa/siRNA SURF6, and control cells was conducted. SURF6 knockdown (5 days post siRNA introduction) causes accumulation of the 47S/45S, 41S pre-rRNA and intermediate18SE. At the same time, pre-rRNAs 30S, 26S, 21S are downmodulated (Fig 4A, 4D and 4G). These data demonstrate that pre-rRNA cleavage that normally occurs at the site 2 is dramatically inhibited in the cells with low level of SURF6. Our findings prove that SURF6 participates in the early steps of 41S pre-rRNA processing and is involved in the processing of 32S precursor.

The observed changes in stationary levels of pre-rRNAs could often be a consequence of several molecular events that could happen at various stages of pre-rRNA processing. Detailed study of molecular events resulting in accumulation/decrease of pre-rRNAs requires analysis of the ratios between several different precursors, the so-called RAMP analysis [19]. RAMP analysis should be performed for every intermediate product of pre-rRNA processing that could be detected by hybridization probes to calculate the ratios in amounts of different precursors (details in Materials and Methods section). Analysis of intensity of bands from Northern blot experiments was performed, and data were presented as RAMP diagrams (Fig 4I). According to RAMP analysis, SURF6 knockdown results in remarkable decrease in 30S/47S ratio. At the same time, 18SE/41S ratio remained stable following SURF6 knockdown in contrast to elevated 18SE/21S. Similar effect could be caused by depletion of factors involved in biogenesis of both LSU and SSU [19, 28]. Lower 30S/47S ratio also points to inhibition of site 2 cleavage. This effect might be caused by more efficient cleavage of E site in the cells with SURF6 deficit.

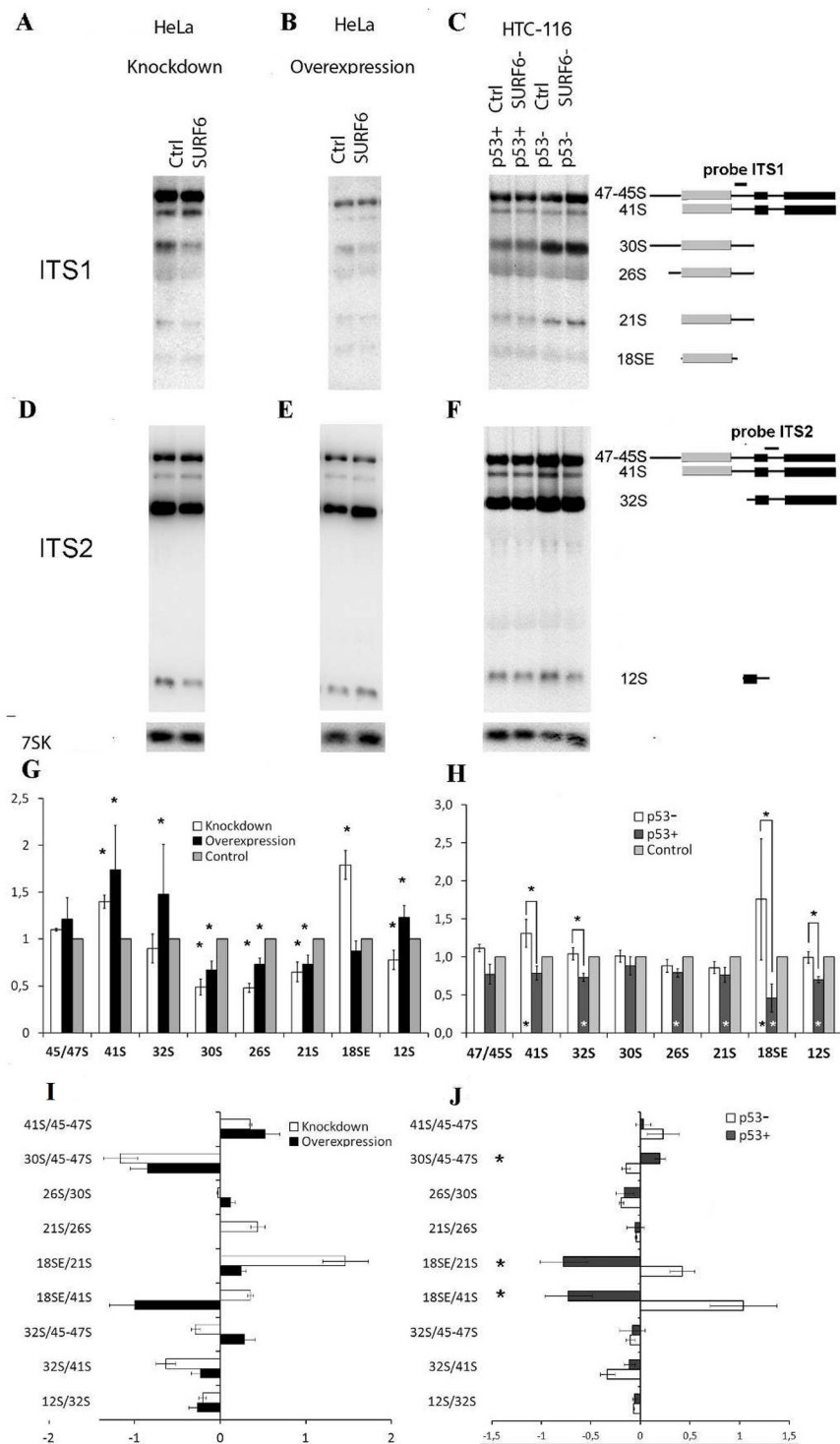

**Fig 4. SURF6 knockdown and overexpression change the profile of pre-rRNA in HeLa cells as well as in HCT116 p53+ and HCT116 p53- cells.** (A-F) Nothern blots of RNA extracts from HeLa, HCT116 p53+ and HCT116 p53- cells with knockdown or overexpression of SURF6 and from control cells. Membranes were hybridized with biotinylated ITS1, ITS2 and 7SK (used for loading control and normalization) probes prior to ECL visualization. Probes hybridization sites on rRNAs are shown at the right side of the panel. (G, H) Quantification of bands intensity corresponding to different precursors in HeLa and HCT116 cells, respectively (results of densitometry analysis). Mean values are shown as bar graphs with standard deviations. Stars indicate statistically significant differences (n = 3). (I, J) Results of the RAMP analysis of changes in pre-rRNA profiles of HeLa and HCT116, respectively. Bars show mean fold

changes (with standard deviations) in the ratios between indicated precursors. Stars indicate statistically significant differences (n = 3, p<0,05). Stars above bars on the panel H indicate statistically significant differences between HCT116 p53+ and p53- cells. Stars at the bottom of the bars show statistically significant differences between either HCT116 p53+ or p53+ and the control (n = 3).

Overexpression of SURF6 also leads to disturbance of the pattern of rRNA processing intermediates. Excessive amounts of SURF6 cause accumulation of the pre-rRNA 32S and 41S as well as 12S, and at the same time leads to a decrease in the content of 30S, 26S, 21S and does not affect the content of 18SE (Fig 4B, 4E and 4G). RAMP analysis shows that SURF6 overexpression results in remarkable decrease in 12S/32S ratio and significantly reduces the ratio of 21S/26S (Fig 4I). Dramatic drop of 12S/32S ratio suggests that SURF6 overexpression leads to inhibition of ITS2 3'site processing. At the same time, 30S/47S, 26S/30S and 21S/26S ratios remained stable following SURF6 overexpression. Thus, overexpression of SURF6 affects pre-rRNA processing along the pathway 1, while pathway 2 remains intact (scheme Fig 3).

## SURF6 knockdown does not slow down proliferation of HeLa cells and increases proportion of cells in S phase

The rate of ribosome biogenesis is tightly linked to proliferation and cell cycle progression. The viability of cells and fractions of cells with different chromatin content was determined by flow cytometry of ethanol-fixed cells stained with the propidium iodide (Fig 5). At 72 hours following administration of RNA duplexes, the proportion of dying cells was approximately two times lower for HeLa with SURF6 knockdown in comparison to control cells with normal SURF6 level. Further analysis of cell cycle using propidium iodide staining demonstrates that in contrast to control, at 48 and 72 hours post siRNA transfection distribution of cells by phases of the cell cycle changes in SURF6 knockdown cells (Fig 5A–5D). Proportion of cells in G1/G0 phase decreases by 6% at 48 hours and by 12% at 72 hours post introduction of siRNA. Proportion of cells in S-phase raised by 6% and 13% at 48 and 72 hours, respectively (Fig 5C and 5D). These unexpected results could be explained by abnormal control and regulation of cell cycle progression in HeLa cells. Surprisingly, overexpression of SURF6 did not result in statistically significant changes in distribution by phases of the cell cycle (Fig 5E–5G). The analysis of cell cycle itself could be misleading without evaluation of cell viability/proliferation. To close this gap, we performed MTT-test to compare viability/proliferation of SURF6 knockdown and control HeLa cells (Fig 5H). In striking contrast to murine fibroblasts, SURF6 knockdown has no negative effect on cell proliferation of HeLa cells.

HeLa cell line has defect in p53 expression [29]. It has been shown that HeLa expresses p53 mRNA, which is not translated, therefore ribosomal stress in HeLa caused by disturbance of pre-rRNA processing possibly would not lead to arrest in G1 phase and subsequent apoptosis. Meanwhile, in human cells with normal p53 expression one would expect negative effect of SURF6 knockdown on cell proliferation and viability.

## Effects of SURF6 knockdown on the pre-rRNA maturation and cell cycle in HCT116 cell line

As it was mentioned before, HeLa cells have defect in p53, meaning that it was not the best choice to study effects of SURF6 on cell cycle. Identification of the potential link between the p53-dependent cell cycle arrest caused by ribosomal stress and cell cycle could be revealed in HCT116 cell sublines which differ only in the expression of the p53 protein (and designated HCT116 p53+ and HCT p53-, respectively). siRNA-mediated knockdown of SURF6 was

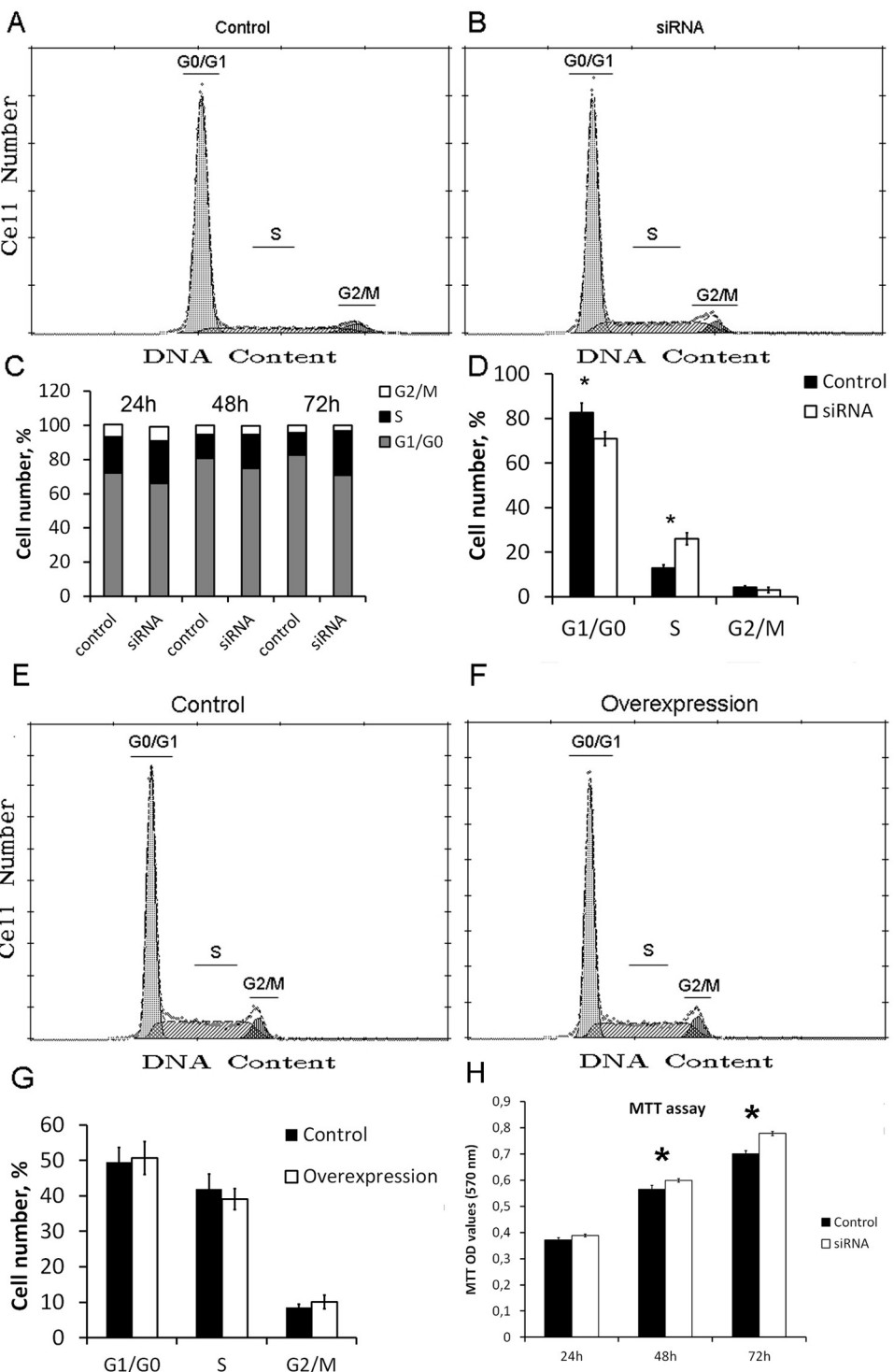

**Fig 5. Effects of SURF6 knockdown and overexpression on the HeLa cell cycle.** (A, B, E, F) Quantitation of DNA content of cells stained with propidium iodide using flow cytometry. Representative histograms show changes in proportions of cells in different stages of the cells cycle in comparison to control cells. (C) Distribution of cells at different stages of the cell cycle at various times after SURF6 knockdown. (D, G) Mean values of proportion of cells in different stages of the cell cycle with standard deviations (n = 3). Stars indicate statistically significant differences between control HeLa and cells with knockdown or overexpression of SURF6 (p<0,05). (H) MTT proliferation assay in HeLa cells with SURF6 knockdown or in control HeLa cells. Data are presented as bar graphs of mean values (n = 3) with standard deviations. Significant difference (p<0,05) is indicated with stars.

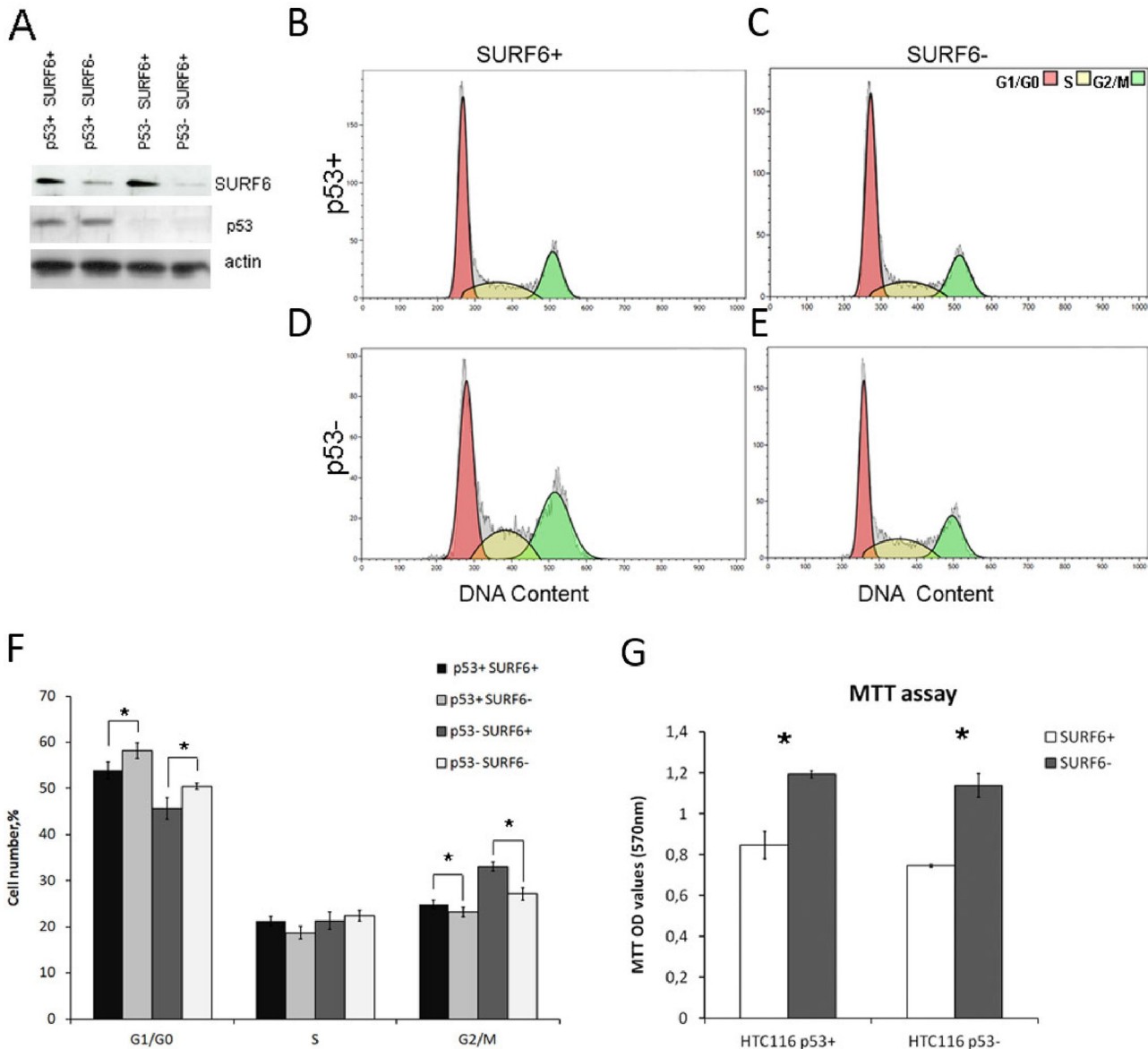

**Fig 6. SURF6 knockdown affects cell cycle distribution of HCT116 human intestinal carcinoma cells.** (A) Western blot analysis of SURF6 and p53 levels in HCT116 p53+ and HCT116 p53- cell lines treated with anti-SURF6 siRNA or non-targeting siRNA as a control. Anti-actin staining was used for normalization. (B) Ratio between p53 and actin in HCT116 p53+ SURF6+ and HCT116 p53+ SURF6- cells. Data presented as mean ± SEM. Statistical analysis for p53 expression corresponds to a biological triplicate. (C-F) Quantitation of DNA content of cells from (A) stained with propidium iodide and analyzed by flow cytometry. Representative histograms show changes in proportions of cells in different stages of the cells cycle in comparison to control cells. (G) Mean percentages of cells in each phase of the cell cycle in four independent experiments. Stars indicate statistically significant differences between control cells HCT116 p53+/SURF6+ or HCT116 p53-/SURF6+, and respective cell lines with siRNA-mediated SURF6 knockdown. (H) MTT assay results obtained in HCT116 p53+/p53- control cells and respective cell lines with SURF6 knockdown and plotted as bar graphs. Mean OD ± SEM values at 570 nm were plotted on the vertical axis. Stars indicate statistically significant differences between control HCT116 and cells with knockdown of SURF6 in their viability/proliferation (p<0,05).

induced in both cell lines by transient transfection with corresponding control and SURF6--specific siRNA (Fig 6A). The accumulation of 41S and 18SE precursors occurs in HCT116 p53- cells similar to effects we observed in HeLa cell (Fig 4G and 4H). At the same time, 32S, 26S, 21S, 18SE and 12S precursors are downregulated in HCT116 p53+ cell line (Fig 4H). This

result suggests that SURF6 knockdown leads to changes in the level of different pre-rRNA precursors depending on the cell line. As mentioned above, the detailed study of changes in pre-rRNA processing requires RAMP analysis. According to RAMP, SURF6 knockdown in HCT116 p53- cell line causes changes in amounts of pre-rRNAs very similar to the that observed in HeLa cells. The most prominent are the drop in 30S/47S ratio along with the elevated 18S/21S ratio (Fig 4J). The opposite effect was found in HCT116 p53+ cells subjected to SURF6 knockdown: 30S/47S ratio was higher, while 18SE/41S and 18SE/21S ratios were decreased in comparison to control cells (Fig 4J). These observations allow assumption that SURF6 knockdown in p53-deficient cells switches the ribosome biogenesis from the pathway 2 to the pathway 1 (Figs 3 and 4). Opposite to this, the shortage of SURF6 in p53-positive cells facilitates the pre-rRNA processing along the pathway 2 while not causing detectable alterations in overall effectiveness of the process. Arguably, the level of p53 protein in HCT116 p53 + could be affected by SURF6 knockdown in control HCT116 p53+ cells. This possibility was ruled out by evaluation of p53 level in HCT116 p53- and p53+ cell lines following SURF6 knockdown (Fig 6A and 6B). One can see that p53 level was not affected by SURF6 knockdown (Fig 6B).

Dependence of effects caused by SURF6 on the ribosomal biogenesis and cell cycle on p53 function could be directly proven by comparison of proliferation/viability of p53+ and p53- cells with SURF6 knockdown. We compared the cell cycle distribution of control and SURF6 knockdown cells expressing or lacking p53. In both p53+ and p53- HCT116 sublines SURF6 knockdown caused statistically significant increase in proportion of cells in G0/G1 phase, had no effect on S phase and markedly decreased G2/M phase (Fig 6C–6G). According to MTT-test, proliferation/survival of HCT116 p53+ or p53- cell lines was not negatively influenced by SURF6 knockdown (Fig 6H).

Based on these data, we suggest SURF6 affecting cell cycle in HCT116 cells largely in p53-independent manner. According to our guess, the presence of p53 allows cells selecting the optimal ribosome biogenesis pathway to evade cell cycle arrest under the SURF6 shortage conditions.

## Discussion

In mammals there are the two main paths of 47S pre-rRNA processing (pathways 1 and 2, see Fig 3) [30, 31]. Both start from the formation of 45S pre-rRNA by removal of 5'-01 and 3'ETS from primary 47S transcript. Then processing follows the pathways 1 or 2 (Fig 3). Progression along both pathways requires participation and precise orchestration of binding and activity of multiple endo- and exonucleases, assembly and dissociation of numerous protein complexes controlling the order and timing of pre-rRNA cleavage events. 3D structures of several presubunit complexes were resolved, many participants of ribosome biogenesis were characterized from functional standpoint in yeast. In this regard, human SURF6 is not an exclusion. It has yeast ortholog Rrp14 which participates in biogenesis of pre-60S and pre-90S in yeast [6]. It is required for polypeptide exit tunnel formation and conversion of pre-rRNA 27S to 25S and 5.8S rRNA by ITS2 cleavage [7] and production of 60S ribosomal subunit [6, 8, 32]. Sequence homology between SURF6 and Rrp14 imply that human SURF6 may have similar role in biogenesis of mammalian ribosomes. Function of murine Surf6 was studied in NIH/3T3 fibroblasts [10, 11]. Zatsepina and colleagues used doxycycline inducible system to elevate Surf6 in 3T3 murine embryonic fibroblasts harboring full length Surf6 cDNA or to downmodulate Surf6 in 3T3 cells by transient transfection of genetic construct harboring antisense DNA [10, 11]. Doxycycline-induced Surf6 overexpression in 3T3 cells resulted in increased proliferation rate and altered pre-rRNA processing. Meanwhile, the antisense-mediated

knockdown of Surf6 led to growth arrest and higher rate of cell death. These results demonstrated involvement of murine Surf6 in ribosome biogenesis without providing detailed molecular mechanism.

It was shown that human SURF6 is associated with proteins EBNA1BP2/Ebp2 and NOP52/Rrp1 [33]. These two proteins are involved in large ribosomal subunit biogenesis [33, 34], although a direct association of trimeric SURF6/Ebp2/NOP52 complex with the pre-60S has not been proven. Human SURF6 can be co-immunoprecipitated or co-isolated with multiple nucleolar proteins including abundant nucleophosmin (B23). Significance of this interaction has not been defined yet. Sequence homology between human SURF6, murine Surf6 and yeast Rrp14 imply that SURF6 and its orthologs may have similar roles in the biogenesis of ribosomes.

Our data suggest that in the absence of functional p53, for instance in human HeLa and HCT116 p53- cells, the shortage of SURF6 forces the usage of the pathway 1 to process pre-rRNA. Both deficit of SURF6 or its overexpression leads to accumulation of 47S/45S and 41S pre-rRNA accompanied by the decrease in 30S and 21S in HeLa cells. Similar, but less profound effects were observed in HCT116 p53- cells with SURF6 siRNA knockdown. (Fig 4A–4J). On the other side, SURF6 knockdown in HCT116 p53+ cells result in opposite effects, and cells switch the ribosome biogenesis to the pathway 2 (Figs 3, 4C, 4F, 4H and 4J). Our data suggest SURF6 involvement in the cleavage at the site 2 of pre-rRNA 47S/45S that initiates the pathway 2 and cleavage at site 2 in the 41S pre-rRNA in the beginning of the pathway 1. This means also that SURF6 is not required for E site cleavage in 41S and/or 21S pre-rRNAs in both pathways. Apart from this, SURF6 function may be supported with other p53-dependent factor or factors with a redundant function, which allow cells to support pathway 2 of the ribosome biogenesis without notable downregulation of the net effectiveness of this process. This hypothesis is supported by decreased level of 30S pre-rRNA along with the elevation of 18SE pre-rRNA (Fig 4A–4J) in SURF6 knockdown p53- HCT116 cells and the opposite effect on pre-rRNA processing in p53+ HCT116 [19]. It is also possible that SURF6 affects the cleavage at the site 4a (see Fig 3). It follows from the notable decrease of 12S pre-rRNA level in HeLa cells with SURF6 knockdown. Accumulation of these RNAs could be also caused by inhibition of cleavage of 41S pre-rRNAs at the site 2 which should happen earlier than at the site 4a (Fig 3). Wang et al. reported the effects caused by depletion of factors involved in SSU and LSU assembly on pre-rRNA processing [19]. In agreement with these findings, one cleavage (out of two) of ITS1 was almost completely inhibited, according to our data, correlating with defects of maturation of the subunit formed from the precursor that is the most adjacent to a cleavage site. It was hypothesized that cleavages of pre-rRNA at the sites A2 and A3 of yeast ITS1 (sites E and 2 in humans) are synchronized with early steps of SSU and LSU assembly (Fig 3). Consequently, each subunit stays attached to the ITS1 until it reaches the maturation step sufficient for cleavage that releases ITS1. The knockdown of human Nop52/Rrp1 similar to SURF6 knockdown, results in accumulation of 41S and 36S pre-rRNA, and lower level of 32S and 12S pre-rRNA [29]. These changes suggest inhibition of the path 1 of pre-rRNA processing. Earlier, it was shown that SURF6 could be co-immunoprecipitated with NOP52 from yeast and human cells [35, 36]. Thus, the deficit of SURF6 may lead to a failure of NOP52 and SURF6 complex formation and deregulation of pre-rRNA processing.

Human 32S precursor is abundant and relatively stable pre-rRNA. It is subjected to splitting cut at the site 4a leading to the formation of 12S and 28S pre-rRNAs. The decrease of the 32S/47S ratio could be interpreted by the shortage of pre-rRNA 32S that enters the path 5.8S/28S. The reason for 32S deficit is the inhibition of early steps of rRNA processing. The ratio 12S/32S that indicates the rate of the next step of 32S processing is increased upon the SURF6 knockdown in HeLa cells. In contrast, the 12S/32S ratio is strongly decreased upon the SURF6

overexpression. This effect of overexpression is likely caused by negative influence of the SURF6 on premature cleavage of the ITS2 at the site 4a. Suppression of the SURF6 also inhibits the cleavage at the site 2, but not at the site E of ITS1. It is possible that interruption of the LSU maturation causes the disbalance and facilitates the incision at the site E. It has been shown earlier that disturbance of the LSU assembly may affect the ITS1 processing at the site A3 in yeast (corresponds to site 2 in humans) [19].

We conclude that SURF6 may be required for efficient cleavage at the site 2, and in the presence of p53 its function may be supported by other factor(s) which may help cell to choose more effective ribosome biogenesis pathway to adapt to SURF6 shortage. The mechanism of pathways switching remains unknown, but the order of cleavage has been shown to vary depending on species, cell type, physiological and developmental stages. Efficiency of the site E cleavage is not markedly affected by SURF6 knockdown. This interpretation follows from ability of HeLa cells to use bypass way of 18SE pre-rRNA generation and skip intermediates 30S, 26S and 20S (see pathway 2, Figs 3 and 4).

Earlier, it has been reported that Surf6 knockdown in MEF 3T3 leads to elevated cell death in less than 2 days post knockdown induction [10]. In contrast to this phenotype, overexpression of Surf6 in MEF did not cause elevated cell death. Here we described the effects of knockdown and overexpression of SURF6 in tumor human cell line HeLa as well as in HCT116 p53 + and HCT116 p53- cells. Overexpression of SURF6 does not result in negative effects on survival and proliferation at 48 hours post transfection. siRNA-mediated knockdown of SURF6 also did not affected viability of neither HeLa nor HCT116 cells at least for 3 days post induction. We also failed to detect visible changes in cell morphology and fibrillar compartment of the nucleoli stained with anti-fibrillarin and anti-SURF6. The same holds true for B23/nucleophosmin which serves as granular nucleolar marker (Fig 1D and 1H).

Study of the changes in cell cycle distribution caused by SURF6 knockdown in HeLa, HCT116 p53- and HCT116 p53+ cells revealed surprising results. Unlike murine fibroblasts or primary human cells, HeLa and both HCT116 did not suffer from G1 phase arrest and subsequent apoptosis. Instead, SURF6 knockdown cells proliferated almost as rapidly as control, and moreover, HeLa had an unexpected increase in S phase (Fig 5A–5D). These "strange" phenotypes could be explained by the defect in p53 expression in HeLa cells [29]. They lack functional p53 protein due to translational incompetence of the p53 mRNA. In this regard, we tested the effect of SURF6 knockdown in variants of HCT116 human cell line one of which has normal regulation and amount of p53, while another was modified to induce p53 knockout [18]. These HCT116 sublines demonstrate almost identical cell cycle distribution profile with mild accumulation of both HCT116 p53+ and p53- cells in G1/G0 phase. MTT assay data obtained in both HCT cell lines correlate well with data obtained in HeLa and demonstrate increased proliferation potential on the third day after SURF6 knockdown.

To our great surprise, the effect of SURF6 on the cell cycle as well as proliferation of cell was independent of the p53 protein expression, but at the same time the pre-rRNA profile and the choice of ribosome biogenesis pathway are influenced by the presence of p53. This suggests that p53-mediated induction of cell growth arrest induced by ribosomal stress might be somehow dependent on the strength of biogenesis impairment, its length and the specificity of depleted factors. Moreover, strong response coupled with cell cycle arrest followed by apoptosis is observed predominantly during inhibition of rDNA transcription, accumulation of free ribosomal proteins or, in very rare cases, during the depletion of ribosome biogenesis factors [37, 38]. Tafforeau et al. performed knockdown of 286 ribosome biogenesis factors and chose 21 of them to evaluate the effects of their depletion in both p53 + and p53- cell lines. They postulate that pre-rRNA profiles observed in knockdown cells do not correlate with the p53 level and represent the earliest events as a response to the

depletion of proteins and on potential changes in p53 level may occur much later [39]. Apart from this, human nucleolar proteome is represented by at least 4500 proteins with sometimes overlapping or redundant functions (in comparison to yeast S. cerevisiae genome database containing only about 303 nucleolar proteins) suggesting that the ribosomal biogenesis in humans could be more flexible. This fact may help explaining the differences between *S. cerevisiae* which undergo cell death following Rrp14 knockdown and human cells which stay proliferative following SURF6 knockdown.

## Supporting information

**S1 Fig. SURF6 co-localizes with ITS2 part of pre-rRNA in the nucleoli of HeLa cells.** The confocal images of the nuclei stained with anti-SURF6 antibody (A, D, G, J; green) or FISH hybridized with the probes specific to ITS1 (B), ITS2 (E), 18S rRNA (H), 28S rRNA (K) (red). Panels (C, F, I, L)–merged images of SURF6 and FISH confocal sections. Arrows point to the areas of nucleoli where the difference between FISH and immunostaining are clearly visible. Representative images are shown. (M) Quantitation of overlap (in %) between SURF6 and FISH presented as bar graph. (N) Structure of human 47S pre-rRNA, red lines correspond to positions of the FISH probes.
(TIF)

**S1 Data.**
(DOCX)

**S1 Raw images.**
(PDF)

## Author Contributions

**Conceptualization:** Anastasiia Moraleva, Maria Kordyukova, Mikhail Polzikov, Olga Zatsepina.

**Data curation:** Anastasiia Moraleva, Maria Kordyukova, Kseniya Shishova, Kira Dobrochaeva.

**Formal analysis:** Anastasiia Moraleva, Olga Zatsepina.

**Funding acquisition:** Maria Kordyukova, Maria Rubtsova, Olga Zatsepina.

**Investigation:** Anastasiia Moraleva, Alexander Deryabin, Maria Kordyukova, Kseniya Shishova, Kira Dobrochaeva.

**Methodology:** Maria Kordyukova, Mikhail Polzikov, Olga Zatsepina.

**Project administration:** Yury Rubtsov, Maria Rubtsova, Olga Dontsova, Olga Zatsepina.

**Resources:** Mikhail Polzikov, Yury Rubtsov, Olga Zatsepina.

**Software:** Anastasiia Moraleva, Alexander Deryabin.

**Supervision:** Yury Rubtsov, Maria Rubtsova, Olga Dontsova, Olga Zatsepina.

**Validation:** Anastasiia Moraleva, Maria Kordyukova, Olga Zatsepina.

**Visualization:** Anastasiia Moraleva, Maria Kordyukova, Kseniya Shishova, Olga Zatsepina.

**Writing – original draft:** Anastasiia Moraleva, Maria Kordyukova, Yury Rubtsov, Maria Rubtsova.

**Writing – review & editing:** Anastasiia Moraleva, Maria Kordyukova, Yury Rubtsov, Maria Rubtsova.

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
