## [Decision Letter · Decision Letter 0]

14 Mar 2022

PONE-D-22-03434Human nucleolar protein SURF6/RRP14 participates in early steps of pre-rRNA processingPLOS ONE

Dear Dr. Moraleva,

Thank you for submitting your manuscript to PLOS ONE. After careful consideration, we feel that it has merit but does not fully meet PLOS ONE’s publication criteria as it currently stands. Both reviewers have done a very good job providing relevant scientific questions that will help the authors to improve their manuscript. I consider that a major revision is fully consistent with Plos One Policy. Therefore, we invite you to submit a revised version of the manuscript that addresses the points raised during the review process.

We look forward to receiving your revised manuscript.

Kind regards,

Jorge Perez-Fernandez, Ph.D.

Academic Editor

PLOS ONE

Journal Requirements:

"This work was financially supported by the Russian Foundation for Basic Research (project no 20-04-00796)."

We note that you have provided funding information. However, funding information should not appear in the Acknowledgments section or other areas of your manuscript. We will only publish funding information present in the Funding Statement section of the online submission form. 

"M Rubtsova

20-04-00796

Russian Foundation for Basic Research 

https://www.rfbr.ru/rffi/eng

NO"

Reviewers' comments:

Reviewer's Responses to Questions

**Comments to the Author**

1. Is the manuscript technically sound, and do the data support the conclusions?

Reviewer #1: Partly

Reviewer #2: Partly

2. Has the statistical analysis been performed appropriately and rigorously? 

Reviewer #1: Yes

Reviewer #2: No

3. Have the authors made all data underlying the findings in their manuscript fully available?

Reviewer #1: Yes

Reviewer #2: Yes

4. Is the manuscript presented in an intelligible fashion and written in standard English?

Reviewer #1: Yes

Reviewer #2: No

5. Review Comments to the Author

Reviewer #1: In their manuscript, the authors characterize the role of Surf6 on cell growth and ribosome biogenesis in the Hela cell line. Surf6 is the homolog of the well-characterized yeast factor Rrp14, which is involved in the early stages of ribosome assembly. The authors and other researchers have studied Surf6 in mice and humans. The results available in the bibliography support a functional conservation between Surf6 and Rrp14, and also indicate that Surf6 is a growth factor. Recently, Surf6 was characterized as a direct partner of Nucleophosmin (NPM1), a component of the nucleolus, suggesting that Surf6 is also present in this sub-nucleolar compartment.

To fully characterize the function of Surf6 in humans, the authors analyzed its localization by immunofluorescence experiments and confirmed its nucleolar accumulation. Then, they studied the effect of its knockdown (KD) and overexpression (OE) on pre-rRNA processing and confirmed a functional conservation between Rrp14 and Surf6 on pre-rRNA processing. Finally, they evaluated the effect of its depletion and overproduction on the cell cycle by flow cytometric analysis.

First of all, I would like to emphasize that the figures are of good quality. However, several points lead to question the overall significance of the work and fail to convince that this manuscript is worth publishing.

My main comments are as follows:

- The authors analyzed the co-localization of Surf6 (by an immuno-fluorescence (IF) approach) with different pre-rRNA precursors (ITS1, ITS2, and 18S) probed by FISH (Figure 1). This method, using fluorescence, is not resolutive enough to conclude on which pre-rRNAs Surf6 is best co-localized. Electron microsocpy would be the only tool to access to such data. Furthermore, the reduced co-localization of Surf6 with 18S and 28S does not make sense since mature 18S and 28S rRNAs accumulate mainly in the cytoplasm. This entire section simply serves to confirm that Surf6 is primarily accumulated in the nucleolus, and is therefore redundant with Figure 2.

- In Figure 2, the authors compare the effect of Surf6 depletion and overexpression on cell morphology. To do so, they perform IF against Surf6 and obtained a nucleolar signal (co-localization with nucleolus marker is missing) that is stronger when Surf6 is overexpressed, but is absent after its depletion, confirming the good depletion conditions. Then, to assess the effect of its EO and KD on cell morphology, they use DAPI staining and phase contrast. These two methods are not appropriate for testing the change in cell morphology associated with ribosomal assembly defects. Indeed, examination of nucleolus staining would be much more informative and the only way to conclude on the effect of alterations in Surf6 expression on cell morphology. I would invite the authors to perform such analysis by probing the nucleolus either by FISH (ITS1) or by using the IF against classical nucleolus markers (NPM1, fibrillarin etc...).

- Finally, the authors analyzed the effect of Surf6 OE and KD on the cell cycle (Figure 4). When ribosome assembly is affected, a nucleolar stress response pathway is activated and leads to p53 accumulation. This generally promotes cell cycle arrest in G1 and even apoptosis if this stress is maintained. Here, in stark contrast to what the authors observe in Figure 3 supporting a ribosome assembly defect, no G1 arrest is observed, only a longer S phase could be noted.

These unusual results should be clarified and discussed in the text. For example, is p53 expression affected under these conditions, we know that lipofectamine transfection clearly plays a role on p53 activation by itself, which could disrupt subsequent cell cycle analysis. Another possibility to interpret this lack of G1 arrest after ribosomal assembly defects would be a direct role of Surf6 in the regulation of p53 in response to nucleolar stress. These results require further scrutiny before publication, as there are many elements that may impact cell cycle analysis.

Minor comments:

-Figure 3, In the RAMP analysis €"18S" should be replaced by 18SE. In the human precursor of 18SE, it is 21S, please also check the labeling in panel (E).

-The comments on Figure 4 are missing in the discussion section.

Overall, I think the manuscript suffers from two major flaws: (1) Lack of nucleolar straining after both Surf6 overexpression and knockdown (2) Necessity to perform additonal controls to properly interpret absence of G1 arrest after Surf6 depletion.

Reviewer #2: In this manuscript, Moraleva et al. investigate the importance of human SURF6 in ribosome biogenesis and cell cycle progression. Using a combination of co-localisation studies, pre-ribosomal RNA analysis and flow cytometry in HeLa cells, they found that knockdown (KD) or overexpression (OE) of SURF6 affects ribosome production (by switching pre-rRNA processing to an alternate pathway) and also has a mild impact on the cell cycle, while, in contrast to previous studies in embryonic fibroblasts, this does not result in cellular death.

While the presented data are of interest, in particular the novel observation that SURF6 might be needed for site 2 pre-rRNA cleavage in human cells, the advances compared to one of their previously published studies in mouse cells are not that obvious (Anastasiia Moraleva, Charalambos Magoulas, Mikhail Polzikov, Sabine Hacot, Hichem C. Mertani, Jean-Jacques Diaz & Olga Zatsepina (2017) Involvement of the specific nucleolar protein SURF6 in regulation of proliferation and ribosome biogenesis in mouse NIH/3T3 fibroblasts, Cell Cycle, 16:20, 1979-1991, DOI: 10.1080/15384101.2017.1371880).

Indeed, it is slightly odd that the authors compare most of their findings with data obtained for the yeast SURF6 homologue (Rrp14), while little reference is made how/if the role of SURF6 is similar/different in humans and mice.

Most experiments are performed to a good standard, but further analysis and/or controls could significantly improve the results and their interpretation:

Figures 1 and 2:

The inclusion of markers such as UBF, Fibrillarin or B23 (see REF 24, Yoshikawa et al.) is needed to define the sub-nucleolar localisation (FC, DFC, GC) of SURF6, which would give further insight into its role(s) in ribosome biogenesis.

While the quantitative analysis of co-localisation of SURF6 with pre-rRNA appears sound (Figure 1), it is not at all clear from the images why co-localisation of SURF6 with ITS2 is different to the other pre-rRNA segments tested?

Likewise, the phase contrast and DAPI is not sufficient to draw comprehensive conclusions about the effect of KD/OE of SURF6 on the cell (Figure 2)– for example, how does KD/OE of SURF6 affect the integrity of the nucleolus and its substructures (see above)?

Figure 3:

While RAMP analysis has been performed to characterise ratios between different precursors (an approach that is well-recognised in the ribosome biogenesis field), the Northern blots presented in panels A and B lack a picture of the mature rRNA levels (e.g. a methylene blue stain of the membrane) or another loading control that is not linked to ribosome biogenesis (e.g. RNase P or 7SK). Levels of the SSU processome snoRNA U3 are included – but this is not a valid loading control, given that U3 levels might change upon KD/OE of SURF6.

How have pre-rRNA levels presented in panel D been normalised? Which statistical test has been performed to define significance?

While Figure 3 contains really interesting results, they need further validation before publication (as described above). In addition, both the figure itself and the description of the results are currently very confusing – as several precursors (in figure and text) and cleavage sites (in text) are mislabelled and/or used incorrectly.

For example:

Line 254/255: “Lower 21S/47S ratio also points to inhibition of site C cleavage.”

- This should be site 2, not site C.

Line 241/242: “It should be pointed that neither SURF6 knockdown, nor SURF6 overexpression resulted in accumulation of aberrant intermediate 34S pre-rRNA variant.”

- 34S is not present anywhere in the figure?

Other specific points that should also be addressed:

Introduction:

- It would be helpful to include the pre-rRNA processing scheme in Figure 1, so that it can be referred to in the introduction.

Methods:

- Please include REF numbers for the SURF6 CDS used for cloning.

- Please include siRNA sequences.

- Please check for spelling – e.g. 1,2% agarose (instead of 1.2%) or 20MM (instead of 20 mM (Line 88).

- What is the sequence of the ITS2 probe used for Northern blotting?

- Where does the ITS1 northern blot probe hybridise with respect to sites E and C?

- Why is cDNA analysis described?

- Please provide sequences for FISH probes.

Results:

- In the description of Figure 2, the text does not match the order in which the panels are described – please double-check, as KD and OE data appear to be mixed up. Which statistical test has been used to test significance in the graphs?

- Please include data (currently not shown) on cell death analysis (Line 231) or perform MTT analysis as in previous publication in mouse cells?

- Given that, as stated in Line 272 “These unexpected results could be explained by abnormal control and regulation of cell cycle progression in HeLa cells” – would it be useful to repeat KD of SURF6 in another human cell line, for example HCT116 cells, which, in contrast to HeLa cells, have a functional p53-signalling pathway?

Discussion:

- Line 304: “It was hypothesized that cleavages of pre-rRNA at the sites 2b and 2c of yeast ITS1 (correspond to Е and С sites in humans) are synchronized with early steps of SSU and LSU assembly (Fig 3C).”

The equivalent yeast sites are called A2 and A3, please correct.

- Line 308: “Earlie, it was shown that SURF6 could be co-immunoprecipitated with NOP52 from yeast and human cells [24].”

I could not find evidence for this in the paper.

- Line 318: “It has been shown earlier that disturbance of the LSU assembly may affect the ITS1 processing at the site 2c (corresponds to site C in humans).”

Please add a reference for this statement.

- Line 320: “We conclude that SURF6 may be required for efficient cleavage at the site C. Efficiency of the site E cleavage is not markedly affected by SURF6 knockdown.”

This is very confusing. The data presented in Figure 3 suggest that SURF6 is needed for site 2 cleavage, and that site E might be used in the absence of SURF6 instead.

Figure 3:

- Please check labelling of precursors in panels A and B (what is 17S, why is 47S not labelled here, but appears in panels D and E).

- Panel C – cleavage site 4a is labelled as E on the very left.

- Panel E – what is 20S? check ratios, e.g. 18S/20S – should this be 18SE/21S instead?

References:

- Not all references contain the journal name, e.g. #12 and #23

- #26 and #28 are the same.

6. PLOS authors have the option to publish the peer review history of their article (what does this mean?). If published, this will include your full peer review and any attached files.

Reviewer #1: No

Reviewer #2: No

---

## [Author Response · Author response to Decision Letter 0]

9 Jun 2022

Rebuttal letter for manuscript

PONE-D-22-03434

Moraleva et al. Human nucleolar protein SURF6/RRP14 participates in early steps of pre-rRNA processing

PLOS ONE

Dear 

Dr. Jorge Perez-Fernandez,

First, my colleagues and I would like to thank you and both reviewers for careful and professional evaluation of our manuscript. We are pleased to receive your kind remarks concerning the quality of data and scientific value of observations described in our study. The issues and concerns raised by the reviewers are reasonable, and it took time and effort to perform some extra experiments and analysis to answer most (in our opinion), if not all, points. The revised version of the manuscript is greatly improved and not only shows the involvement of SURF6 to processing of pre-rRNA, but also provides hints concerning the changes in cell cycle caused by overexpression or downmodulation of SURF6 in human cells.

Please, find our point-by-point response to reviewers below.

Reviewers’ comments are shown in italic. Our response is in straight font. Changes in the main text of the manuscript are labeled yellow. 

Reviewer #1: In their manuscript, the authors characterize the role of Surf6 on cell growth and ribosome biogenesis in the Hela cell line. Surf6 is the homolog of the well-characterized yeast factor Rrp14, which is involved in the early stages of ribosome assembly. The authors and other researchers have studied Surf6 in mice and humans. The results available in the bibliography support a functional conservation between Surf6 and Rrp14, and also indicate that Surf6 is a growth factor. Recently, Surf6 was characterized as a direct partner of Nucleophosmin (NPM1), a component of the nucleolus, suggesting that Surf6 is also present in this sub-nucleolar compartment.

To fully characterize the function of Surf6 in humans, the authors analyzed its localization by immunofluorescence experiments and confirmed its nucleolar accumulation. Then, they studied the effect of its knockdown (KD) and overexpression (OE) on pre-rRNA processing and confirmed a functional conservation between Rrp14 and Surf6 on pre-rRNA processing. Finally, they evaluated the effect of its depletion and overproduction on the cell cycle by flow cytometric analysis.

First of all, I would like to emphasize that the figures are of good quality. However, several points lead to question the overall significance of the work and fail to convince that this manuscript is worth publishing.

We appreciate the effort made by the reviewer to judge our study. And happy to provide detailed response to his comments:

My main comments are as follows:

- The authors analyzed the co-localization of Surf6 (by an immuno-fluorescence (IF) approach) with different pre-rRNA precursors (ITS1, ITS2, and 18S) probed by FISH (Figure 1). This method, using fluorescence, is not resolutive enough to conclude on which pre-rRNAs Surf6 is best co-localized. Electron microsocpy would be the only tool to access to such data. Furthermore, the reduced co-localization of Surf6 with 18S and 28S does not make sense since mature 18S and 28S rRNAs accumulate mainly in the cytoplasm. This entire section simply serves to confirm that Surf6 is primarily accumulated in the nucleolus, and is therefore redundant with Figure 2.

We agree with the reviewer that combination (overlay) of FISH for ITS1, ITS2, 18S and 28S probes to pre-rRNA fragments and IF for SURF6 is not the best way to answer the question concerning co-localization of SURF6 with the regions of nucleolus involved in biogenesis of rRNA precursors. We had in mind that we will be not get definitive answers concerning precise co-localization of SURF6 with areas where particular rRNA precursors reside. But these data gave us a clue and indicated that in normal HeLa cells SURF6 localization significantly overlaps with areas where these precursors could be found. Therefore, Fig. 1 shows simply that it’s reasonable to imply that SURF6 in HeLa cells participates in early steps of the rRNA biogenesis. Concerning 18S and 28S, it is true that majority of these RNAs are localized in the cytoplasm, but in Fig.1 we show their localization as parts of larger precursors. We changed Fig. 1D to make clearer where corresponding FISH probes anneal in full length pre-rRNA. 

Figure 2 was changed significantly (see figures and figure legends) by adding confocal microscopy of cells co-stained with SURF6 and nucleolar markers that label functional zones of nucleolus, fibrillarin and B23. Therefore, Fig. 1 and new Fig. 2 lack redundancy noted by reviewer. 

Significant part of the text was modified and is shown below (lines 249 – 257 of revised text):

In our previous study we demonstrated that murine Surf6 is involved in ribosome biogenesis, which was not shown for human SURF6. Therefore, we decided to study co-localization of SURF6 with ITS1, ITS2, 18S and 28S parts of pre-rRNA in nucleoli of fixed/permeabilized HeLa cells using FISH with respective specific probes (Fig. 1N, scheme of 47S pre-rRNA and location of probes) and immunostaining of SURF6 (Fig. 1). The goal of this experiment was to determine at what stage (early or late) of pre-rRNA processing SURF6 might participate. Despite an imperfection of this approach, it is clear from confocal images that signal from SURF6 almost completely overlaps inside the nucleoli with the signal from ITS2 FISH (Fig. 1D, E, F). At the same time, SURF6 also significantly co-localizes with the areas of ITS1 FISH signal (about 70% overlap, Fig. 1A, B, C). Overlay of SURF6 immunostaining with FISH of 18S and 28S was less prominent (Fig. 1G-L).

- In Figure 2, the authors compare the effect of Surf6 depletion and overexpression on cell morphology. To do so, they perform IF against Surf6 and obtained a nucleolar signal (co-localization with nucleolus marker is missing) that is stronger when Surf6 is overexpressed, but is absent after its depletion, confirming the good depletion conditions. Then, to assess the effect of its EO and KD on cell morphology, they use DAPI staining and phase contrast. These two methods are not appropriate for testing the change in cell morphology associated with ribosomal assembly defects. Indeed, examination of nucleolus staining would be much more informative and the only way to conclude on the effect of alterations in Surf6 expression on cell morphology. I would invite the authors to perform such analysis by probing the nucleolus either by FISH (ITS1) or by using the IF against classical nucleolus markers (NPM1, fibrillarin etc...).

We would like to agree with the concern of the reviewer. To solve the issue, we performed confocal microscopy of fixed control HeLa cells and cells with SURF6 knockdown that were triple stained with DAPI, anti-SURF6, and anti-fibrillarin (fibrillar nucleolar component marker) or anti-B23 (granular nucleolar component marker). One can see (Fig. 2, D, H, N, R) that morphology of fibrillar component and global architecture of the nucleolus is not affected by the SURF6 KD. 

Following text was added to the manuscript (lines 280-285 of revised text):

To examine potential changes in fine architecture of nucleolus, we co-stained nucleoli of control HeLa and SURF6 knockdown HeLa with anti-SURF6 and anti-fibrillarin (nucleolar fibrillar component marker) or anti-SURF6 and anti-B23 (nucleolar granular component marker) (Fig. 2C-J, M-T). Confocal images demonstrate that fine patterning of fibrillarin and B23 are not affected by SURF6 knockdown. It is interesting that fibrillarin signal is more punctate than that of SURF6 suggesting that SURF6 could potentially function outside fibrillar centers.

In case of overexpression, we transfected HeLa cells with SURF6-GFP fusion construct and stained them with DAPI and the probe to ITS1. Corresponding supplemental figure (see Suppl. Fig. 2) demonstrates that morphology of the nucleolus is not visibly affected by overexpression of SURF6-GFP chimera.

Please, see changes in the text (lines 295-300 of revised text):

Further, we stained cells that overexpress SURF6-GFP fusion with ITS1 FISH probe to demonstrate that fine structure of nucleolus is not affected as well (Suppl. Fig. 2). These results suppose that overexpression of human SURF6 in HeLa does not disturb cell morphology similar to the consequence of SURF6 overexpression in mouse embryonic cells [11]. Formally, it is not possible to exclude that there are faint changes in nucleoli of HeLa cells following overexpression or knockdown of SURF6, but they would require complicated high resolution EM studies.

- Finally, the authors analyzed the effect of Surf6 OE and KD on the cell cycle (Figure 4). When ribosome assembly is affected, a nucleolar stress response pathway is activated and leads to p53 accumulation. This generally promotes cell cycle arrest in G1 and even apoptosis if this stress is maintained. Here, in stark contrast to what the authors observe in Figure 3 supporting a ribosome assembly defect, no G1 arrest is observed, only a longer S phase could be noted.

These unusual results should be clarified and discussed in the text. For example, is p53 expression affected under these conditions, we know that lipofectamine transfection clearly plays a role on p53 activation by itself, which could disrupt subsequent cell cycle analysis. Another possibility to interpret this lack of G1 arrest after ribosomal assembly defects would be a direct role of Surf6 in the regulation of p53 in response to nucleolar stress. These results require further scrutiny before publication, as there are many elements that may impact cell cycle analysis.

Again, we would like to thank the reviewer for asking this important question. 

First, for HeLa cell cycle in SURF6 WT/KD, we added MTT test data that show statistically insignificant but detectable increase in viability of SURF6 KD cells in comparison to WT. This indicates, although not formally proves, that SURF6 KD does not decrease survival of cells.

Please, see text (lines 380-383 of revised text):

The analysis of cell cycle itself could be misleading without evaluation of cell viability/proliferation. To close this gap, we performed MTT-test to compare viability/proliferation of SURF6 knockdown and control HeLa cells (Fig. 4H). In striking contrast to murine fibroblasts, SURF6 knockdown has no negative effect on cell proliferation of HeLa cells. 

Second, for some reason HeLa cells have very short G2 phase (Poudyal D, Herman A, Adelsberger JW, Yang J, Hu X, Chen Q, Bosche M, Sherman BT, Imamichi T. A novel microRNA, hsa-miR-6852 differentially regulated by Interleukin-27 induces necrosis in cervical cancer cells by downregulating the FoxM1 expression. Sci Rep. 2018 Jan 17;8(1):900. doi: 10.1038/s41598-018-19259-4.) and lack expression of functional p53 protein (doi:10.1101/gad.9.17.2170Genes & Dev. 1995. 9: 2170-2183).

Therefore, we tend to speculate that this abnormal cell cycle and loss of control from p53 in WT HeLa affects the changes in cell cycle distribution of HeLa with SURF6 KD/OE. It should be also stressed out that despite very significant changes in pre-rRNA processing we do not see decreased viability of cells by MTT. 

Corresponding text (lines 384-387 of revised text):

HeLa cell line has defect in p53 expression [26]. It has been shown that they express p53 mRNA, which is not translated, therefore ribosomal stress in HeLa caused by disturbance of pre-rRNA processing possibly would not lead to arrest in G1 phase and subsequent apoptosis. Meanwhile, in human cells with normal p53 expression one would expect negative effect of SURF6 knockdown on cell proliferation and viability.

Third, according to the literature, there is a link between p53 and ribosome biogenesis/ribosomal stress. In human cancer cells the control of cell cycle from p53 is often lost which means that data obtained in cancer cell lines could be affected by this defect. To address this point, we set up SURF6 siRNA transient KD experiments in HCT116 p53+ and HCT116 p53- cell lines. HCT116 p53+ cell line has normal regulation of p53 activity. The results of cell cycle distribution and MTT test are presented in Fig.5 which we added to the manuscript. Fig.5A demonstrates efficiency of siRNA mediated KD in both cell lines. Analysis of cell cycle distribution (Fig.5B, C) clearly indicates that, independent on the presence of p53, SURF6 KD increases proportion of cells in G0/G1 and decreases in G2/M phase. S phase does not change significantly. This result is in the striking contrast with HeLa cells where we see effect on S phase. Viability of cells in both p53- and p53+ cells go up in the case of SURF6 KD (Fig.5D). From these results we concluded that SURF6-mediated effects on cell cycle are independent on p53, despite dramatic effects on pre-rRNA processing. This raises the possibility that SURF6 may have two distinct functions, or that somehow deficiency in SURF6 and ribosome biogenesis defects are uncoupled from the p53 regulation of cell cycle in HCT116 cells. This unexpected phenomenon deserves further study.

We added section to the manuscript (lines 388-399 of revised text):

SURF6 knockdown affects the cell cycle in HCT116 cell line in p53-independent manner

The question regarding a role of p53 in SURF6 effects on the cell cycle could be answered by direct comparison of cell cycle and proliferation/viability of p53 sufficient and p53 deficient cells with SURF6 knockdown. We used a pair of HCT116 cell lines which differ only in expression of the p53 protein (and designated HCT116 p53+ and HCT p53-, respectively). siRNA-mediated knockdown of SURF6 was induced in both cell lines by transient transfection with corresponding control and SURF6-specific siRNA (Fig. 5A). Then we compared the cell cycle distribution of control and SURF6 knockdown cells expressing or lacking p53. In both p53+ and p53- HCT116 SURF6 knockdown caused statistically significant increase in proportion of cells in G0/G1 phase, had no effect on S phase and markedly decreased G2/M phase (Fig. 5B-F). According to MTT-test, proliferation/survival of HCT116 p53+ or p53- cell lines was not negatively influenced by SURF6 knockdown (Fig. 5G). Based on this data, we suggest that SURF6 affects cell cycle in HCT116 cell line largely in p53-independent manner. 

Minor comments:

-Figure 3, In the RAMP analysis €"18S" should be replaced by 18SE. In the human precursor of 18SE, it is 21S, please also check the labeling in panel (E).

We thank reviewer for pointing to the mistake.

Changes were made in the Fig. 3E, 18S was replaced by 18SE.

-The comments on Figure 4 are missing in the discussion section.

We thank the reviewer for this comment. The discussion section was updated according to the data from figures 4 and 5 (lines 455-461 of revised text):

Study of cell cycle distribution changes caused by SURF6 knockdown in HeLa revealed surprising results. Unlike murine fibroblasts or primary human cells, HeLa did not suffer from G1 phase arrest and subsequent apoptosis. Instead, they proliferated as good as control HeLa and had unexpected increase in S phase (Fig. 4A-D). These strange phenotypes could be explained by the defect in p53 expression in HeLa cells [26]. They lack functional p53 protein due to translational incompetence of the p53 mRNA. In this regard, we tested the effect of SURF6 knockdown in variants of HCT116 human cell line one of which has normal regulation and amount of p53, while another was modified to induce p53 knockout [13].

Overall, I think the manuscript suffers from two major flaws: (1) Lack of nucleolar straining after both Surf6 overexpression and knockdown (2) Necessity to perform additonal controls to properly interpret absence of G1 arrest after Surf6 depletion.

We fixed both major flaws by making additional experiments and providing data (please, see response to major points and Fig. 2D, H, N, R, Suppl. Fig. 2, Fig. 4H, and Fig.5). Text of the manuscript was changed accordingly.

Reviewer #2: In this manuscript, Moraleva et al. investigate the importance of human SURF6 in ribosome biogenesis and cell cycle progression. Using a combination of co-localisation studies, pre-ribosomal RNA analysis and flow cytometry in HeLa cells, they found that knockdown (KD) or overexpression (OE) of SURF6 affects ribosome production (by switching pre-rRNA processing to an alternate pathway) and also has a mild impact on the cell cycle, while, in contrast to previous studies in embryonic fibroblasts, this does not result in cellular death.

While the presented data are of interest, in particular the novel observation that SURF6 might be needed for site 2 pre-rRNA cleavage in human cells, the advances compared to one of their previously published studies in mouse cells are not that obvious (Anastasiia Moraleva, Charalambos Magoulas, Mikhail Polzikov, Sabine Hacot, Hichem C. Mertani, Jean-Jacques Diaz & Olga Zatsepina (2017) Involvement of the specific nucleolar protein SURF6 in regulation of proliferation and ribosome biogenesis in mouse NIH/3T3 fibroblasts, Cell Cycle, 16:20, 1979-1991, DOI: 10.1080/15384101.2017.1371880).

Indeed, it is slightly odd that the authors compare most of their findings with data obtained for the yeast SURF6 homologue (Rrp14), while little reference is made how/if the role of SURF6 is similar/different in humans and mice.

We would like to thank reviewer for his time and outstanding expertise.

There are reasons why we avoided the bold comparisons between the effects of SURF6 KD/OE on ribosomal biogenesis in HeLa cells and murine fibroblasts. First, nucleolus architecture in humans and rodents are quite different, despite evolutionary closeness. Second, the effect of SURF6 KD on proliferation and survival of murine and human HeLa cells is opposite. It is likely linked to the lack of p53 which normally arrests cell cycle in response to ribosomal stress. 

Most experiments are performed to a good standard, but further analysis and/or controls could significantly improve the results and their interpretation.

Figures 1 and 2:

The inclusion of markers such as UBF, Fibrillarin or B23 (see REF 24, Yoshikawa et al.) is needed to define the sub-nucleolar localisation (FC, DFC, GC) of SURF6, which would give further insight into its role(s) in ribosome biogenesis.

As it was mentioned in the response to the first reviewer’s comments, we performed co-staining of HeLa nucleoli with anti SURF6 and either anti-fibrillarin or anti-B23 (Fig.2 D, H, N, R). Confocal imaging shows that SURF6 pretty much co-localizes with granular nucleolar component marked by B23, while fibrillarin signal that marks FC covers smaller areas. This possibly indicates involvement of SURF6 in early, but not the earliest stages of ribosomal biogenesis.

The following text was added to the manuscript (lines 280-285 and 295-300 of revised text):

To examine potential changes in fine architecture of nucleolus, we co-stained nucleoli of control HeLa and SURF6 knockdown HeLa with anti-SURF6 and anti-fibrillarin (nucleolar fibrillar component marker) or anti-SURF6 and anti-B23 (nucleolar granular component marker) (Fig. 2C-J, M-T). Confocal images demonstrate that fine patterning of fibrillarin and B23 are not affected by SURF6 knockdown. It is interesting that fibrillarin signal is more punctate than that of SURF6 suggesting that SURF6 could potentially function outside fibrillar centers.

Further, we stained cells that overexpress SURF6-GFP fusion with ITS1 FISH probe to demonstrate that fine structure of nucleolus is not affected as well (Suppl. Fig. 2). These results suppose that overexpression of human SURF6 in HeLa does not disturb cell morphology similar to the consequence of SURF6 overexpression in mouse embryonic cells [11]. Formally, it is not possible to exclude that there are faint changes in nucleoli of HeLa cells following overexpression or knockdown of SURF6, but they would require complicated high resolution EM studies.

While the quantitative analysis of co-localisation of SURF6 with pre-rRNA appears sound (Figure 1), it is not at all clear from the images why co-localisation of SURF6 with ITS2 is different to the other pre-rRNA segments tested?

We agree with the reviewer that pictures of ITS2 staining look somewhat different. Quantitation and analysis using ImageJ software tool show that statistically they are very similar to results obtained with other pre-rRNA probes. 

Likewise, the phase contrast and DAPI is not sufficient to draw comprehensive conclusions about the effect of KD/OE of SURF6 on the cell (Figure 2)– for example, how does KD/OE of SURF6 affect the integrity of the nucleolus and its substructures (see above)?

This issue was addressed by providing extra data presented in Fig. 2 (C-J, M-T) and Suppl. Fig. 2. Nucleoli of HeLa WT/KD were stained with anti-fibrillarin or B23 and SURF6 to demonstrate preserved architecture of this compartment. One can see that distribution of fibrillarin or B23 look very similar in the case of normal or downmodulated level of SURF6 in HeLa. For SURF6 OE we transfected HeLa cells with construct coding SURF6-GFP fusion and performed FISH with ITS1 probe (Suppl. Fig. 2). We failed to detect visible defects in nucleoli according to this staining. 

The following text was added (lines 280-285 and 295-300 of revised text):

To examine potential changes in fine architecture of nucleolus, we co-stained nucleoli of control HeLa and SURF6 knockdown HeLa with anti-SURF6 and anti-fibrillarin (nucleolar fibrillar component marker) or anti-SURF6 and anti-B23 (nucleolar granular component marker) (Fig. 2C-J, M-T). Confocal images demonstrate that fine patterning of fibrillarin and B23 are not affected by SURF6 knockdown. It is interesting that fibrillarin signal is more punctate than that of SURF6 suggesting that SURF6 could potentially function outside fibrillar centers.

Further, we stained cells that overexpress SURF6-GFP fusion with ITS1 FISH probe to demonstrate that fine structure of nucleolus is not affected as well (Suppl. Fig. 2). These results suppose that overexpression of human SURF6 in HeLa does not disturb cell morphology similar to the consequence of SURF6 overexpression in mouse embryonic cells [11]. Formally, it is not possible to exclude that there are faint changes in nucleoli of HeLa cells following overexpression or knockdown of SURF6, but they would require complicated high resolution EM studies.

Figure 3:

While RAMP analysis has been performed to characterise ratios between different precursors (an approach that is well-recognised in the ribosome biogenesis field), the Northern blots presented in panels A and B lack a picture of the mature rRNA levels (e.g. a methylene blue stain of the membrane) or another loading control that is not linked to ribosome biogenesis (e.g. RNase P or 7SK). Levels of the SSU processome snoRNA U3 are included – but this is not a valid loading control, given that U3 levels might change upon KD/OE of SURF6.

We agree that U3 is not the best loading control since it’s level in some cases may be affected by changes in ribosome biogenesis. Therefore, we added pictures of ethidium bromide-stained RNA agarose gels used for Northern hybridization analysis (Fig.3 A). These images show that according to signals from 28S and 18S the amount of RNA used for Northern in SURF6 WT and SURF6 KD/OE HeLa cells was almost the same. It points that U3 snoRNA normalization we used is valid. 

How have pre-rRNA levels presented in panel D been normalised? Which statistical test has been performed to define significance?

In panel D pre-rRNA levels were normalized to U3 snoRNA which, according to loading control, is not affected by changes in SURF6 level. Then the levels of corresponding precursors in control (scrambled siRNA-transfected) HeLa cells was considered as 1 and fold changes were plotted as histograms. Student’s t-test was used for statistical evaluation.

While Figure 3 contains really interesting results, they need further validation before publication (as described above). In addition, both the figure itself and the description of the results are currently very confusing – as several precursors (in figure and text) and cleavage sites (in text) are mislabelled and/or used incorrectly.

As it was mentioned before, we added another level of validation using comparison of signals from 28S/18S RNA as loading control. These data demonstrate that U3 snoRNA is not affected by SURF6 KD/OE in HeLa and can be used for normalization. 

Considering mentioned mistakes in Fig.3 and text, they were corrected accordingly (see text and corrected figures in the revised version).

For example:

Line 254/255: “Lower 21S/47S ratio also points to inhibition of site C cleavage.”

- This should be site 2, not site C.

Changed to:

Lower 21S/47S ratio also points to inhibition of site 2 cleavage.

Line 241/242: “It should be pointed that neither SURF6 knockdown, nor SURF6 overexpression resulted in accumulation of aberrant intermediate 34S pre-rRNA variant.”

- 34S is not present anywhere in the figure?

This sentence was deleted from the text since the statement adds nothing to the story.

Other specific points that should also be addressed.

Introduction:

- It would be helpful to include the pre-rRNA processing scheme in Figure 1, so that it can be referred to in the introduction.

We would like to thank the reviewer for helpful suggestion. 

We changed Fig.1 to make it easier for understanding by modifying panel D. Labels were added for all probes (ITS1/2, 18S, 28S) and their positioning was shown in bold red lines. Separate labels for ITS1 and ITS2 were added above pre-rRNA scheme.

Then, we examined the introduction and found that pre-rRNA processing and steps of this process are mentioned only twice in introduction. Removal of pre-rRNA processing scheme from Fig.3 would make it very hard for understanding. At the same time, addition of Fig.3C to Fig.1 would make it overcrowded. Using this reasoning, while feeling apologetic for arguing the reviewer, we still decided to keep initial order of figures.

Methods:

- Please include REF numbers for the SURF6 CDS used for cloning.

Corresponding RefSeq NM_006753.6 was added to the text.

- Please include siRNA sequences.

Anti-SURF6 and control siRNA duplexes (Life technologies, USA; anti-SURF6 siRNA id S13655, neg. control id am4635).

- Please check for spelling – e.g. 1,2% agarose (instead of 1.2%) or 20MM (instead of 20 mM (Line 88).

Thank you for finding these errors. The text was edited accordingly.

- What is the sequence of the ITS2 probe used for Northern blotting?

The sequence of the ITS2 probe was added to Materials and Methods, please, see section Nothern blotting.

- Where does the ITS1 northern blot probe hybridise with respect to sites E and C? According the comparison of ITS1 segment and ITS1 probe sequences it needs to be clarified that ITS1 probe don’t match neither E nor C sites.

Changes were made in Fig.3C to clarify the position of the ITS1 probe. It does not overlap with sites E or C.

- Why is cDNA analysis described?

We apologize for this error. We deleted the section considering cDNA synthesis since we used previously cloned cDNA to make genetic constructs mentioned in the manuscript.

- Please provide sequences for FISH probes.

Sequences of the probes were added to the Materials and Methods.

Results:

- In the description of Figure 2, the text does not match the order in which the panels are described – please double-check, as KD and OE data appear to be mixed up. Which statistical test has been used to test significance in the graphs?

Thank you for correction. We carefully checked text in the legend and corresponding figure. Please, see edited legend to Fig.2. We used Student’s t-test to confirm statistical significance.

- Please include data (currently not shown) on cell death analysis (Line 231) or perform MTT analysis as in previous publication in mouse cells?

Results of MTT analysis performed on HeLa (Fig.4H) and HCT116 p53+/p53- (Fig. 5D) cells with SURF6 KD and control cells. Following description was added to the main text (lines 380-383 and 397-399 in revised text):

The analysis of cell cycle itself could be misleading without evaluation of cell viability/proliferation. To close this gap, we performed MTT-test to compare viability/proliferation of SURF6 knockdown and control HeLa cells (Fig.4H). In striking contrast to murine fibroblasts, SURF6 knockdown has no negative effect on cell proliferation of HeLa cells. 

According to MTT-test, proliferation/survival of HCT116 p53+ or p53- cell lines was not negatively influenced by SURF6 knockdown (Fig. 5G). Based on this data, we suggest that SURF6 affects cell cycle in HCT116 cell line largely in p53-independent manner. 

- Given that, as stated in Line 272 “These unexpected results could be explained by abnormal control and regulation of cell cycle progression in HeLa cells” – would it be useful to repeat KD of SURF6 in another human cell line, for example HCT116 cells, which, in contrast to HeLa cells, have a functional p53-signalling pathway?

We conducted the suggested experiment in HCT116 p53+ human tumor cell line and its variant with p53 KO – HCT116 p53-. KD of SURF6 in these cell lines led to elongation of G0/G1 and shortening of G2/M (S was not affected). This finding suggests that SURF6 KD influences cell cycle independently of p53. It also supports our speculation regarding abnormal cell cycle in HeLa. Main text was chanced accordingly, please, see below. We added significant chunk of text as well (lines 388-399 of revised text):

SURF6 knockdown affects the cell cycle in HCT116 cell line in p53-independent manner

The question regarding a role of p53 in SURF6 effects on the cell cycle could be answered by direct comparison of cell cycle and proliferation/viability of p53 sufficient and p53 deficient cells with SURF6 knockdown. We used a pair of HCT116 cell lines which differ only in expression of the p53 protein (and designated HCT116 p53+ and HCT p53-, respectively). siRNA-mediated knockdown of SURF6 was induced in both cell lines by transient transfection with corresponding control and SURF6-specific siRNA (Fig. 5A). Then we compared the cell cycle distribution of control and SURF6 knockdown cells expressing or lacking p53. In both p53+ and p53- HCT116 SURF6 knockdown caused statistically significant increase in proportion of cells in G0/G1 phase, had no effect on S phase and markedly decreased G2/M phase (Fig. 5B-F). According to MTT-test, proliferation/survival of HCT116 p53+ or p53- cell lines was not negatively influenced by SURF6 knockdown (Fig. 5G). Based on this data, we suggest that SURF6 affects cell cycle in HCT116 cell line largely in p53-independent manner.

Discussion:

- Line 304: “It was hypothesized that cleavages of pre-rRNA at the sites 2b and 2c of yeast ITS1 (correspond to Е and С sites in humans) are synchronized with early steps of SSU and LSU assembly (Fig 3C).”

The equivalent yeast sites are called A2 and A3, please correct.

Corrected:

It was hypothesized that cleavages of pre-rRNA at the sites A2 and A3 of yeast ITS1 (correspond to Е and С sites in humans) are synchronized with early steps of SSU and LSU assembly

- Line 308: “Earlie, it was shown that SURF6 could be co-immunoprecipitated with NOP52 from yeast and human cells [24].”

I could not find evidence for this in the paper.

We thank the reviewer for the detailed analysis of presented data. This link was changed for the proper one (Horsey EW, Jakovljevic J, Miles TD, Harnpicharnchai P, Woolford JL. Role of the yeast Rrp1 protein in the dynamics of pre-ribosome maturation. RNA. 2004;10: 813–827. doi:10.1261/rna.5255804).

- Line 318: “It has been shown earlier that disturbance of the LSU assembly may affect the ITS1 processing at the site 2c (corresponds to site C in humans).”

Please add a reference for this statement.

Presented data is shown in the following paper: Wang M, Anikin L, Pestov DG. Two orthogonal cleavages separate subunit RNAs in mouse ribosome biogenesis. 2014;42: 11180–11191. doi:10.1093/nar/gku787. It’s reference #14 in the text.

- Line 320: “We conclude that SURF6 may be required for efficient cleavage at the site C. Efficiency of the site E cleavage is not markedly affected by SURF6 knockdown.”

This is very confusing. The data presented in Figure 3 suggest that SURF6 is needed for site 2 cleavage, and that site E might be used in the absence of SURF6 instead.

We agree that the interpretation that we provided is confusing. We changed the main text accordingly:

The data presented in Figure 3 suggest that SURF6 is needed for site 2 cleavage, and that site E might be used in the absence of SURF6 instead.

Figure 3:

- Please check labelling of precursors in panels A and B (what is 17S, why is 47S not labelled here, but appears in panels D and E).

The following reference to the article that shows detection of 17S:

Morello LG, Hesling C, Coltri PP, Castilho BA, Rimokh R, Zanchin NI. The NIP7 protein is required for accurate pre-rRNA processing in human cells. Nucleic Acids Res. 2011 Jan;39(2):648-65. doi: 10.1093/nar/gkq758. Epub 2010 Aug 26. PMID: 20798176; PMCID: PMC3025556.

Correct labeling of the 45S/45S doublet bands was added to Fig. 3A.

- Panel C – cleavage site 4a is labelled as E on the very left.

Corrected, please, see Fig. 3C.

- Panel E – what is 20S? check ratios, e.g. 18S/20S – should this be 18SE/21S instead?

Corrected, please, see Fig. 3C

- Not all references contain the journal name, e.g. #12 and #23

- #26 and #28 are the same.

Please, see corrected list of references.

---

## [Decision Letter · Decision Letter 1]

11 Jul 2022

PONE-D-22-03434R1Human nucleolar protein SURF6/RRP14 participates in early steps of pre-rRNA processingPLOS ONE

Dear Dr. Moraleva,

Thank you for submitting your manuscript to PLOS ONE. After careful consideration, we feel that it has merit but does not fully meet PLOS ONE’s publication criteria as it currently stands. Therefore, we invite you to submit a revised version of the manuscript that addresses the points raised during the review process. In this case, reviewer 2 still raises many concerns which require further attention.  Although a major revision in a second round would suggest manuscript rejection, the substantial changes in the manuscript changed the suggestion of reviewer 1 towards minor revision. I consider, the authors made a strong effort in improving the manuscript. Therefore, it is worth a third round.Please include the following items when submitting your revised manuscript:A rebuttal letter that responds to each point raised by the academic editor and reviewer(s). You should upload this letter as a separate file labeled 'Response to Reviewers'.A marked-up copy of your manuscript that highlights changes made to the original version. You should upload this as a separate file labeled 'Revised Manuscript with Track Changes'.An unmarked version of your revised paper without tracked changes. You should upload this as a separate file labeled 'Manuscript'.Please submit your revised manuscript by Aug 25 2022 11:59PM. If you will need more time than this to complete your revisions, please reply to this message or contact the journal office at plosone@plos.org. If applicable, we recommend that you deposit your laboratory protocols in protocols.io to enhance the reproducibility of your results. Protocols.io assigns your protocol its own identifier (DOI) so that it can be cited independently in the future. For instructions see: https://journals.plos.org/plosone/s/submission-guidelines#loc-laboratory-protocols. Additionally, PLOS ONE offers an option for publishing peer-reviewed Lab Protocol articles, which describe protocols hosted on protocols.io. Read more information on sharing protocols at https://plos.org/protocols?utm_medium=editorial-email&utm_source=authorletters&utm_campaign=protocols.

We look forward to receiving your revised manuscript.

Kind regards,

Jorge Perez-Fernandez, Ph.D.

Academic Editor

PLOS ONE

Reviewers' comments:

Reviewer's Responses to Questions

**Comments to the Author**

1. If the authors have adequately addressed your comments raised in a previous round of review and you feel that this manuscript is now acceptable for publication, you may indicate that here to bypass the “Comments to the Author” section, enter your conflict of interest statement in the “Confidential to Editor” section, and submit your "Accept" recommendation.

Reviewer #1: All comments have been addressed

Reviewer #2: (No Response)

2. Is the manuscript technically sound, and do the data support the conclusions?

Reviewer #1: Yes

Reviewer #2: Partly

3. Has the statistical analysis been performed appropriately and rigorously? 

Reviewer #1: Yes

Reviewer #2: Yes

4. Have the authors made all data underlying the findings in their manuscript fully available?

Reviewer #1: Yes

Reviewer #2: Yes

5. Is the manuscript presented in an intelligible fashion and written in standard English?

Reviewer #1: Yes

Reviewer #2: Yes

6. Review Comments to the Author

Reviewer #1: In the present manuscript, the authors have presented a more comprehensive version of the functional study of Surf6. To test the function of Surf6 in Hela cells, the authors used two complementary approaches, studying the effect of depletion (via a siRNAs strategy) or overexpression of this factor on nucleolus integrity, pre-rRNA processing, and the cell cycle. With the addition of several experiments using new nucleolar markers for nucleolus staining, additional cell lines (HCT-116 p53+ and p53-) and MTT assays, the authors have fully addressed my main comments. Nevertheless, it remains quite intriguing that no proliferation defects are observed following Surf6 depletion in human Hela and HCT-116 cell lines, whereas homologs of this factor are essential in yeast and mouse.

The authors discuss this observation in the last section of the “Discussion”section, and propose that this could be due to a potential role of Surf6 "in the p53-mediated growth arrest cascade". Nevertheless, the observation that mature 18S and 25S rRNA levels are not affected after Surf6 depletion argues for other hypotheses:

1- For some reason, Surf6 depletion is not strong enough to completely impair pre-rRNA processing, although 90% of the depletion observed by the authors should be sufficient to completely impair ribosome production.

2- In these cells, Surf6 function could also be supported by an additional factor with a redundant function.

3- In these cells, switching from pathway 1 to pathway 2 during pre-rRNA processing does not alter the efficiency of ribosome synthesis.

I think the authors could also insert these concepts into their discussion.

Reviewer #2: The revised manuscript by Moraleva et al. is an improvement on the previous version, which addresses many major points raised, for example with respect to the subnucleolar localisation of SURF6, but some of the other points/newly included experiments require further clarification before publication.

Major concern:

The presented experiments in HeLa and HCT116 cells reveal that the effect of SURF6 KD on cell proliferation is the opposite in human and murine cells. The authors mention that this is likely linked to the lack of p53 in HeLa cells, which normally arrests the cell cycle in response to ribosomal stress. However, while this explanation makes sense for HeLa cells, it does not explain why HCT116 cells are different from murine cells (which express p53 as well), especially given the new data presented in Figure 5 saying that the effect of SURF6 KD on the cell cycle is p53 independent in HCT116 cells.

As it stands, the manuscript provides pre-rRNA processing data for HeLa cells, which are not a suitable model for cell cycle analysis. On the other hand, Figure 5 shows interesting cell cycle data for HCT116 p53+ cells – but not enough information to explain how SURF6 levels impact on ribosome assembly/p53 levels/cell cycle/survival etc.

It would therefore be essential to check if SURF6 KD or OE in HCT116 p53+ cells indeed leads to changes in pre-rRNA processing and, as a consequence, p53 levels – which would be expected as the nucleolar stress response often seen upon ribosome assembly defects.

Other points:

• Figure 1: In agreement with reviewer 1, I also feel that Figure 1 is redundant given that much better data is provided in Figure 2.

• Figure 2 and Supplementary Figure 2: the co-localisation of SURF6 with Fibrillarin and B23 presented in Figure 2 is really informative. Why was a different, less informative method (i.e. ITS1 FISH) used to assess nucleolar structure upon SURF6 OE (Supplementary Figure 2), especially given that the figure does not show a direct comparison of control vs OE cells? It would be beneficial to analyse Fibrillarin and B23 staining upon SURF6 OE vs control cells.

• Figure 3: the quality of the newly included EtBr-stained gel to show levels of mature 28S and 18S rRNA is poor and not suitable to allow quantitative analysis of the U3 snoRNA, which is then used as a standard to normalise RNA levels. This experiment requires a suitable loading control for normalisation - either an RNA not involved in ribosome assembly, such as the RNA component of RNase P, as mentioned previously, or, at the very least, suitable assessment of mature 28S and 18S rRNA levels by Northern blotting.

Page 2, line 59:

“In contrast to primates, nucleolus in mice, despite evolutionary closeness of rodents and primates, lacks important structures.”

- This statement requires a reference.

Page 12, line 345:

“utilization of E site is prevented by SURF6 depletion”

- This should say: utilization of E site is promoted by SURF6 depletion”

Page 12, line 353, 367, 368:

“It was hypothesized that cleavages of pre-rRNA at the sites A2 and A3 of yeast ITS1 (correspond to Е and С sites in humans)”

“Suppression of the SURF6 also inhibits the cleavage at the site C, but not at the site E of ITS1.”

“It has been shown earlier that disturbance of the LSU assembly may affect the ITS1 processing at the site 2c (corresponds to site C in humans) [14].”

- Please replace “site C” with “site 2”.

Legend to Figure 5, panel H:

“Stars indicate statistically significant differences between control HeLa and cells with knockdown of SURF6 in their viability/proliferation.”

- These are HCT116, not HeLa cells.

7. PLOS authors have the option to publish the peer review history of their article (what does this mean?). If published, this will include your full peer review and any attached files.

Reviewer #1: No

Reviewer #2: No

---

## [Author Response · Author response to Decision Letter 1]

10 Jan 2023

Reviewer #1: In the present manuscript, the authors have presented a more comprehensive version of the functional study of Surf6. To test the function of Surf6 in Hela cells, the authors used two complementary approaches, studying the effect of depletion (via a siRNAs strategy) or overexpression of this factor on nucleolus integrity, pre-rRNA processing, and the cell cycle. With the addition of several experiments using new nucleolar markers for nucleolus staining, additional cell lines (HCT-116 p53+ and p53-) and MTT assays, the authors have fully addressed my main comments. Nevertheless, it remains quite intriguing that no proliferation defects are observed following Surf6 depletion in human Hela and HCT-116 cell lines, whereas homologs of this factor are essential in yeast and mouse.

We thank the reviewer for this essential comment. Indeed, human SURF6 has close homologs in yeast and mice. It should be noted that despite evolutionary conservation, the organization of nucleus and smaller structures, especially nucleolus, are quite different between primitive single cell eukaryotes (yeast) and higher vertebrates such as primates and rodents (human and mouse). Thus, at least for yeast and human, we can guess that difference in effects of SURF6 knockdown or overexpression on proliferation could be explained by higher complexity of ribosomal biogenesis in humans and, possibly, development of redundant pathways, which could compensate the loss of one, or several factors involved in the pre-rRNA processing and assembly of subunits. However, this is not the case for human and mouse. 

Regarding current knowledge of SURF6 role in ribosomal biogenesis, most of information was obtained in experiments with yeast homologue of SURF6 – Rrp14. Rrp14 was found to be associated with pre-60S and pre-90S ribosomal precursors [Yamada H et al. Yeast Rrp14p is a nucleolar protein involved in both ribosome biogenesis and cell polarity. RNA. 2007;13: 1977–87]. It was shown that Rrp14 is required for conversion of pre-rRNA 27S to 25S and 5.8S rRNA by ITS2 cleavage [Oeffinger M et al. Yeast Rrp14 is required for ribosomal subunit synthesis and for correct positioning of the mitotic spindle during mitosis. Nucleic Acids Res. 2007;35: 1354–66] and participates in production of 60S ribosomal subunit [Yamada H et al. RNA. 2007;13: 1977–87; Sanghai ZA et al. Modular assembly of the nucleolar pre-60S ribosomal subunit. Nature. 2018;556: 126–129; Chaker-Margot M, Klinge S. Assembly and early maturation of large subunit precursors. RNA. 2019;25: 465–471]. Human SURF6 is associated with proteins EBNA1BP2/Ebp2 and NOP52/Rrp1 [Yoshikawa H et al. Human nucleolar protein Nop52 (RRP1/NNP-1) is involved in site 2 cleavage in internal transcribed spacer 1 of pre-rRNAs at early stages of ribosome biogenesis. Nucleic Acids Res. 2015;43: 5524–5536]. Both proteins are involved in large ribosomal subunit (LSU) biogenesis [Romanova L et al. Critical Role of Nucleostemin in Pre-rRNA Processing. 2008; Yoshikawa H et al. Nucleic Acids Res. 2015;43: 5524–5536], although a direct association of this trimeric complex with the pre-60S need to be proven. Sequence homology between SURF6 and Rpr14 imply that SURF6 may have similar role in biogenesis of mammalian ribosomes. At the same time, there is a lack of information regarding molecular mechanisms that link the process of ribosomal biogenesis with the control of the cell cycle in higher eukaryotes. This makes any parallels between yeast and vertebrates speculative. 

Concerning mouse and humans, we still could not rule out that difference in effects of SURF6 knockdown on cell proliferation may strongly depend on the choice of the cell line. We use immortalized or transformed cells, which usually have defects in control of the cell cycle, which could be related or independent of the p53 function. Another explanation is that rodents and primates have differences in the pathways of rRNA biogenesis. 

We added substantial pieces of text to Discussion section to explain the discrepancy noted by the reviewer.

In mammals there are the two main paths of 47S pre-rRNA processing (pathways 1 and 2, see Fig. 3) [30, 31]. Both start from the formation of 45S pre-rRNA by removal of 5’-01 and 3’ETS from primary 47S transcript. Then processing follows the pathways 1 or 2 (Fig. 3). Progression along both pathways requires participation and precise orchestration of binding and activity of multiple endo- and exonucleases, assembly and dissociation of numerous protein complexes controlling the order and timing of pre-rRNA cleavage events. 3D structures of several pre-subunit complexes were resolved, many participants of ribosome biogenesis were characterized from functional standpoint in yeast. In this regard, human SURF6 is not an exclusion. It has yeast ortholog Rrp14 which participates in biogenesis of pre-60S and pre-90S in yeast [6]. It is required for polypeptide exit tunnel formation and conversion of pre-rRNA 27S to 25S and 5.8S rRNA by ITS2 cleavage [7] and production of 60S ribosomal subunit [6,8,34]. Sequence homology between SURF6 and Rpr14 imply that human SURF6 may have similar role in biogenesis of mammalian ribosomes. Function of murine SURF6 was studied in NIH/3T3 fibroblasts [10, 11]. Zatsepina and colleagues used doxycycline inducible system to elevate Surf6 in 3T3 murine embryonic fibroblasts harboring full length Surf6 cDNA or to downmodulate Surf6 in 3T3 cells by transient transfection of genetic construct harboring antisense DNA [10, 11]. Doxycycline-induced Surf6 overexpression in 3T3 cells resulted in increased proliferation rate and altered pre-rRNA processing. Meanwhile, the antisense-mediated knockdown of Surf6 led to growth arrest and higher rate of cell death. These results demonstrated involvement of murine Surf6 in ribosome biogenesis without providing detailed molecular mechanism.

It was shown that human SURF6 is associated with proteins EBNA1BP2/Ebp2 and NOP52/Rrp1 [32]. These two proteins are involved in large ribosomal subunit biogenesis [32,33], although a direct association of trimeric SURF6/Ebp2/NOP52 complex with the pre-60S has not been proven. Human SURF6 can be co-immunoprecipitated or co-isolated with multiple nucleolar proteins including abundant nucleophosmin (B23). Significance of this interaction has not been defined yet. Sequence homology between human SURF6, murine Surf6 and yeast Rpr14 imply that SURF6 and its orthologs may have similar roles in the biogenesis of ribosomes.

Our data suggest that in the absence of functional p53, for instance in human HeLa and HCT116 p53- cells, the shortage of SURF6 forces the usage of the pathway 1 to process pre-rRNA. Both deficit of SURF6 or its overexpression leads to accumulation of 47S/45S and 41S pre-rRNA accompanied by the decrease in 30S and 21S in HeLa cells. Similar, but less profound effects were observed in HCT116 p53- cells with SURF6 siRNA knockdown. (Fig. 4A-K). On the other side, SURF6 knockdown in HCT116 p53+ cells result in opposite effects, and cells switch the ribosome biogenesis to the pathway 2 (Fig. 3, fig. 4C, F, H, K). Our data suggest SURF6 involvement in the cleavage at the site 2 of pre-rRNA 47S/45S that initiates the pathway 2 and cleavage at site 2 in the 41S pre-rRNA in the beginning of the pathway 1. This means also that SURF6 is not required for E site cleavage in 41S and/or 21S pre-rRNAs in both pathways. Apart from this, SURF6 function may be supported with other p53-dependent factor or factors with a redundant function, which allow cells to support pathway 2 of the ribosome biogenesis without notable downregulation of the net effectiveness of this process. This hypothesis is supported by decreased level of 30S pre-rRNA along with the elevation of 18SE pre-rRNA (Fig. 4A-K) in SURF6 knockdown p53- HCT116 cells and the opposite effect on pre-rRNA processing in p53+ HCT116 [19]. It is also possible that SURF6 affects the cleavage at the site 4a (see Fig. 3). It follows from the notable decrease of 12S pre-rRNA level in HeLa cells with SURF6 knockdown. Accumulation of these RNAs could be also caused by inhibition of cleavage of 41S pre-rRNAs at the site 2 which should happen earlier than at the site 4a (Fig. 3). Wang et al. reported the effects caused by depletion of factors involved in SSU and LSU assembly on pre-rRNA processing [19]. In agreement with these findings, one cleavage (out of two) of ITS1 was almost completely inhibited, according to our data, correlating with defects of maturation of the subunit formed from the precursor that is the most adjacent to a cleavage site. It was hypothesized that cleavages of pre-rRNA at the sites A2 and A3 of yeast ITS1 (sites E and 2 in humans) are synchronized with early steps of SSU and LSU assembly (Fig. 3). Consequently, each subunit stays attached to the ITS1 until it reaches the maturation step sufficient for cleavage that releases ITS1. The knockdown of human Nop52/Rrp1 similar to SURF6 knockdown, results in accumulation of 41S and 36S pre-rRNA, and lower level of 32S and 12S pre-rRNA [29]. These changes suggest inhibition of the path 1 of pre-rRNA processing. Earlier, it was shown that SURF6 could be co-immunoprecipitated with NOP52 from yeast and human cells [35,36]. Thus, the deficit of SURF6 may lead to a failure of NOP52 and SURF6 complex formation and deregulation of pre-rRNA processing.

//

Earlier, it has been reported that SURF6 knockdown in MEF 3T3 leads to elevated cell death in less than 2 days post knockdown induction [10]. In contrast to this phenotype, overexpression of SURF6 in MEF did not cause elevated cell death. Here we described the effects of knockdown and overexpression of SURF6 in tumor human cell line HeLa as well as in HCT116 p53+ and HCT116 p53- cells. Overexpression of SURF6 does not result in negative effects on survival and proliferation at 48 hours post transfection. siRNA-mediated knockdown of SURF6 also did not affected viability of neither HeLa nor HCT116 cells at least for 3 days post induction. We also failed to detect visible changes in cell morphology and fibrillar compartment of the nucleoli stained with anti-fibrillarin and anti-SURF6. The same holds true for B23/nucleophosmin which serves as granular nucleolar marker (Fig. 1D, H).

Study of the changes in cell cycle distribution caused by SURF6 knockdown in HeLa, HCT116 p53- and HCT116 p53+ cells revealed surprising results. Unlike murine fibroblasts or primary human cells, HeLa and both HCT116 did not suffer from G1 phase arrest and subsequent apoptosis. Instead, SURF6 knockdown cells proliferated almost as rapidly as control, and moreover, HeLa had an unexpected increase in S phase (Fig. 5A-D). These “strange” phenotypes could be explained by the defect in p53 expression in HeLa cells [29]. They lack functional p53 protein due to translational incompetence of the p53 mRNA. In this regard, we tested the effect of SURF6 knockdown in variants of HCT116 human cell line one of which has normal regulation and amount of p53, while another was modified to induce p53 knockout [18]. These HCT116 sublines demonstrate almost identical cell cycle distribution profile with mild accumulation of both HCT116 p53+ and p53- cells in G1/G0 phase. MTT assay data obtained in both HCT cell lines correlate well with data obtained in HeLa and demonstrate increased proliferation potential on the third day after SURF6 knockdown.

The authors discuss this observation in the last section of the “Discussion”section, and propose that this could be due to a potential role of Surf6 "in the p53-mediated growth arrest cascade". Nevertheless, the observation that mature 18S and 25S rRNA levels are not affected after Surf6 depletion argues for other hypotheses:

1-For some reason, Surf6 depletion is not strong enough to completely impair pre-rRNA processing, although 90% of the depletion observed by the authors should be sufficient to completely impair ribosome production.

2-In these cells, Surf6 function could also be supported by an additional factor with a redundant function.

3-In these cells, switching from pathway 1 to pathway 2 during pre-rRNA processing does not alter the efficiency of ribosome synthesis.

I think the authors could also insert these concepts into their discussion.

We are grateful to the reviewer for these interesting suggestions and added required concepts to the Discussion section. It should be stressed out that our new data allow making the conclusion that the 3rd scenario could be the likely explanation of the subtle effects on the ribosomal biogenesis efficiency and, hence, on the cell cycle progression. Please see the Discussion section and the text below: 

In mammals there are the two main paths of 47S pre-rRNA processing (pathways 1 and 2, see Fig. 3) [30, 31]. Both start from the formation of 45S pre-rRNA by removal of 5’-01 and 3’ETS from primary 47S transcript. Then processing follows the pathways 1 or 2 (Fig. 3). Progression along both pathways requires participation and precise orchestration of binding and activity of multiple endo- and exonucleases, assembly and dissociation of numerous protein complexes controlling the order and timing of pre-rRNA cleavage events. 3D structures of several pre-subunit complexes were resolved, many participants of ribosome biogenesis were characterized from functional standpoint in yeast. In this regard, human SURF6 is not an exclusion. It has yeast ortholog Rrp14 which participates in biogenesis of pre-60S and pre-90S in yeast [6]. It is required for polypeptide exit tunnel formation and conversion of pre-rRNA 27S to 25S and 5.8S rRNA by ITS2 cleavage [7] and production of 60S ribosomal subunit [6,8,34]. Sequence homology between SURF6 and Rpr14 imply that human SURF6 may have similar role in biogenesis of mammalian ribosomes. Function of murine SURF6 was studied in NIH/3T3 fibroblasts [10, 11]. Zatsepina and colleagues used doxycycline inducible system to elevate Surf6 in 3T3 murine embryonic fibroblasts harboring full length Surf6 cDNA or to downmodulate Surf6 in 3T3 cells by transient transfection of genetic construct harboring antisense DNA [10, 11]. Doxycycline-induced Surf6 overexpression in 3T3 cells resulted in increased proliferation rate and altered pre-rRNA processing. Meanwhile, the antisense-mediated knockdown of Surf6 led to growth arrest and higher rate of cell death. These results demonstrated involvement of murine Surf6 in ribosome biogenesis without providing detailed molecular mechanism.

It was shown that human SURF6 is associated with proteins EBNA1BP2/Ebp2 and NOP52/Rrp1 [32]. These two proteins are involved in large ribosomal subunit biogenesis [32,33], although a direct association of trimeric SURF6/Ebp2/NOP52 complex with the pre-60S has not been proven. Human SURF6 can be co-immunoprecipitated or co-isolated with multiple nucleolar proteins including abundant nucleophosmin (B23). Significance of these interaction has not been defined yet. Sequence homology between human SURF6, murine Surf6 and yeast Rpr14 imply that SURF6 and its orthologs may have similar roles in the biogenesis of ribosomes.

Our data suggest that in the absence of functional p53, for instance in human HeLa and HCT116 p53- cells, the shortage of SURF6 forces the usage of the pathway 1 to process pre-rRNA. Both deficit of SURF6 or its overexpression leads to accumulation of 47S/45S and 41S pre-rRNA accompanied by the decrease in 30S and 21S in HeLa cells. Similar, but less profound effects were observed in HCT116 p53- cells with SURF6 siRNA knockdown. (Fig. 4A-K). On the other side, SURF6 knockdown in HCT116 p53+ cells result in opposite effects, and cells switch the ribosome biogenesis to the pathway 2 (Fig. 3, fig. 4C, F, H, K). Our data suggest SURF6 involvement in the cleavage at the site 2 of pre-rRNA 47S/45S that initiates the pathway 2 and cleavage at site 2 in the 41S pre-rRNA in the beginning of the pathway 1. This means also that SURF6 is not required for E site cleavage in 41S and/or 21S pre-rRNAs in both pathways. Apart from this, SURF6 function may be supported with other p53-dependent factor or factors with a redundant function, which allow cells to support pathway 2 of the ribosome biogenesis without notable downregulation of the net effectiveness of this process. This hypothesis is supported by decreased level of 30S pre-rRNA along with the elevation of 18SE pre-rRNA (Fig. 4A-K) in SURF6 knockdown p53- HCT116 cells and the opposite effect on pre-rRNA processing in p53+ HCT116 [19]. It is also possible that SURF6 affects the cleavage at the site 4a (see Fig. 3). It follows from the notable decrease of 12S pre-rRNA level in HeLa cells with SURF6 knockdown. Accumulation of these RNAs could be also caused by inhibition of cleavage of 41S pre-rRNAs at the site 2 which should happen earlier than at the site 4a (Fig. 3). Wang et al. reported the effects caused by depletion of factors involved in SSU and LSU assembly on pre-rRNA processing [19]. In agreement with these findings, one cleavage (out of two) of ITS1 was almost completely inhibited, according to our data, correlating with defects of maturation of the subunit formed from the precursor that is the most adjacent to a cleavage site. It was hypothesized that cleavages of pre-rRNA at the sites A2 and A3 of yeast ITS1 (sites E and 2 in humans) are synchronized with early steps of SSU and LSU assembly (Fig. 3). Consequently, each subunit stays attached to the ITS1 until it reaches the maturation step sufficient for cleavage that releases ITS1. The knockdown of human Nop52/Rrp1 similar to SURF6 knockdown, results in accumulation of 41S and 36S pre-rRNA, and lower level of 32S and 12S pre-rRNA [29]. These changes suggest inhibition of the path 1 of pre-rRNA processing. Earlier, it was shown that SURF6 could be co-immunoprecipitated with NOP52 from yeast and human cells [35,36]. Thus, the deficit of SURF6 may lead to a failure of NOP52 and SURF6 complex formation and deregulation of pre-rRNA processing.

Human 32S precursor is abundant and relatively stable pre-rRNA. It is subjected to splitting cut at the site 4a leading to the formation of 12S and 28S pre-rRNAs. The decrease of the 32S/47S ratio could be interpreted by the shortage of pre-rRNA 32S that enters the path 5.8S/28S. The reason for 32S deficit is the inhibition of early steps of rRNA processing. The ratio 12S/32S that indicates the rate of the next step of 32S processing is increased upon the SURF6 knockdown in HeLa cells. In contrast, the 12S/32S ratio is strongly decreased upon the SURF6 overexpression. This effect of overexpression is likely caused by negative influence of the SURF6 on premature cleavage of the ITS2 at the site 4a. Suppression of the SURF6 also inhibits the cleavage at the site 2, but not at the site E of ITS1. It is possible that interruption of the LSU maturation causes the disbalance and facilitates the incision at the site E. It has been shown earlier that disturbance of the LSU assembly may affect the ITS1 processing at the site 2c in yeast (corresponds to site 2 in humans) [19].

We conclude that SURF6 may be required for efficient cleavage at the site 2, and in the presence of p53 its function may be supported by other factor(s) which may help cell to choose more effective ribosome biogenesis pathway to adapt to SURF6 shortage. The mechanism of pathways switching remains unknown, but the order of cleavage has been shown to vary depending on species, cell type, physiological and developmental stages. Efficiency of the site E cleavage is not markedly affected by SURF6 knockdown. This interpretation follows from ability of HeLa cells to use bypass way of 18SE pre-rRNA generation and skip intermediates 30S, 26S and 20S (see pathway 2, Fig. 3, Fig. 4). 

Earlier, it has been reported that SURF6 knockdown in MEF 3T3 leads to elevated cell death in less than 2 days post knockdown induction [10]. In contrast to this phenotype, overexpression of SURF6 in MEF did not cause elevated cell death. Here we described the effects of knockdown and overexpression of SURF6 in tumor human cell line HeLa as well as in HCT116 p53+ and HCT116 p53- cells. Overexpression of SURF6 does not result in negative effects on survival and proliferation at 48 hours post transfection. siRNA-mediated knockdown of SURF6 also did not affected viability of neither HeLa nor HCT116 cells at least for 3 days post induction. We also failed to detect visible changes in cell morphology and fibrillar compartment of the nucleoli stained with anti-fibrillarin and anti-SURF6. The same holds true for B23/nucleophosmin which serves as granular nucleolar marker (Fig. 1D, H).

Study of the changes in cell cycle distribution caused by SURF6 knockdown in HeLa, HCT116 p53- and HCT116 p53+ cells revealed surprising results. Unlike murine fibroblasts or primary human cells, HeLa and both HCT116 did not suffer from G1 phase arrest and subsequent apoptosis. Instead, SURF6 knockdown cells proliferated almost as rapidly as control, and moreover, HeLa had an unexpected increase in S phase (Fig. 5A-D). These “strange” phenotypes could be explained by the defect in p53 expression in HeLa cells [29]. They lack functional p53 protein due to translational incompetence of the p53 mRNA. In this regard, we tested the effect of SURF6 knockdown in variants of HCT116 human cell line one of which has normal regulation and amount of p53, while another was modified to induce p53 knockout [18]. These HCT116 sublines demonstrate almost identical cell cycle distribution profile with mild accumulation of both HCT116 p53+ and p53- cells in G1/G0 phase. MTT assay data obtained in both HCT cell lines correlate well with data obtained in HeLa and demonstrate increased proliferation potential on the third day after SURF6 knockdown.

To our great surprise, the effect of SURF6 on the cell cycle as well as proliferation of cell was independent of the p53 protein expression, but at the same time the pre-rRNA profile and the choice of ribosome biogenesis pathway are influenced by the presence of p53. This suggests that p53-mediated induction of cell growth arrest induced by ribosomal stress might be somehow dependent on the strength of biogenesis impairment, its length and the specificity of depleted factors. Moreover, strong response coupled with cell cycle arrest followed by apoptosis is observed predominantly during inhibition of rDNA transcription, accumulation of free ribosomal proteins or, in very rare cases, during the depletion of ribosome biogenesis factors [37, 38]. Tafforeau et al. performed knockdown of 286 ribosome biogenesis factors and chose 21 of them to evaluate the effects of their depletion in both p53+ and p53- cell lines. They postulate that pre-rRNA profiles observed in knockdown cells do not correlate with the p53 level and represent the earliest events as a response to the depletion of proteins and on potential changes in p53 level may occur much later [39]. Apart from this, human nucleolar proteome is represented by at least 4500 proteins with sometimes overlapping or redundant functions (in comparison to yeast S. cerevisiae genome database containing only about 303 nucleolar proteins) suggesting that the ribosomal biogenesis in humans could be more flexible. This fact may help explaining the differences between S. cerevisiae which undergo cell death following RRP14 knockdown and human cells which stay proliferative following SURF6 knockdown.

Reviewer #2: The revised manuscript by Moraleva et al. is an improvement on the previous version, which addresses many major points raised, for example with respect to the subnucleolar localisation of SURF6, but some of the other points/newly included experiments require further clarification before publication.

Major concern:

The presented experiments in HeLa and HCT116 cells reveal that the effect of SURF6 KD on cell proliferation is the opposite in human and murine cells. The authors mention that this is likely linked to the lack of p53 in HeLa cells, which normally arrests the cell cycle in response to ribosomal stress. However, while this explanation makes sense for HeLa cells, it does not explain why HCT116 cells are different from murine cells (which express p53 as well), especially given the new data presented in Figure 5 saying that the effect of SURF6 KD on the cell cycle is p53 independent in HCT116 cells.

We would like to thank Reviewer for his time and outstanding expertise. Northern blots and followed RAMP analysis were performed to evaluate the effect of the SURF6 knockdown on the pre-rRNA processing in HCT116 p53+ and HCT116 p53- cells. Following description was added to the main text (lines 322-349 of revised text).

As it was mentioned before, HeLa cells have defect in p53, meaning that it was not the best choice to study effects of SURF6 on cell cycle. Identification of the potential link between the p53-dependent cell cycle arrest caused by ribosomal stress and cell cycle could be revealed in HCT116 cell sublines which differ only in the expression of the p53 protein (and designated HCT116 p53+ and HCT p53-, respectively). siRNA-mediated knockdown of SURF6 was induced in both cell lines by transient transfection with corresponding control and SURF6-specific siRNA (Fig. 6A). The accumulation of 41S and 18SE precursors occurs in HCT116 p53- cells similar to effects we observed in HeLa cell (Fig. 4G, H). At the same time, 32S, 26S, 21S, 18SE and 12S precursors are downregulated in HCT116 p53+ cell line (Fig. 4H). This result suggests that SURF6 knockdown leads to changes in the level of different pre-rRNA precursors depending on the cell line. As mentioned above, the detailed study of changes in pre-rRNA processing requires RAMP analysis. According to RAMP, SURF6 knockdown in HCT116 p53- cell line causes changes in amounts of pre-rRNAs very similar to the that observed in HeLa cells. The most prominent are the drop in 30S/47S ratio along with the elevated 18S/21S ratio (Fig. 4K). The opposite effect was found in HCT116 p53+ cells subjected to SURF6 knockdown: 30S/47S ratio was higher, while 18SE/41S and 18SE/21S ratios were decreased in comparison to control cells (Fig. 4K). These observations allow assumption that SURF6 knockdown in p53-deficient cells switches the ribosome biogenesis from the pathway 2 to the pathway 1 (Fig. 3 and Fig.4). Opposite to this, the shortage of SURF6 in p53-positive cells facilitates the pre-rRNA processing along the pathway 2 while not causing detectable alterations in overall effectiveness of the process. Arguably, the level of p53 protein in HCT116 p53+ could be affected by SURF6 knockdown in control HCT116 p53+ cells. This possibility was ruled out by evaluation of p53 level in HCT116 p53- and p53+ cell lines following SURF6 knockdown (Fig.6A). One can see that p53 level was not affected by SURF6 knockdown.

Dependence of effects caused by SURF6 on the ribosomal biogenesis and cell cycle on p53 function could be directly proven by comparison of proliferation/viability of p53+ and p53- cells with SURF6 knockdown. We compared the cell cycle distribution of control and SURF6 knockdown cells expressing or lacking p53. In both p53+ and p53- HCT116 SURF6 knockdown caused statistically significant increase in proportion of cells in G0/G1 phase, had no effect on S phase and markedly decreased G2/M phase (Fig. 6B-F). According to MTT-test, proliferation/survival of HCT116 p53+ or p53- cell lines was not negatively influenced by SURF6 knockdown (Fig. 6G). 

Based on these data, we suggest SURF6 affecting cell cycle in HCT116 cells largely in p53-independent manner. According to our guess, the presence of p53 allows cells selecting the optimal ribosome biogenesis pathway to evade cell cycle arrest under the SURF6 shortage conditions.

As it stands, the manuscript provides pre-rRNA processing data for HeLa cells, which are not a suitable model for cell cycle analysis. On the other hand, Figure 5 shows interesting cell cycle data for HCT116 p53+ cells – but not enough information to explain how SURF6 levels impact on ribosome assembly/p53 levels/cell cycle/survival etc.

It would therefore be essential to check if SURF6 KD or OE in HCT116 p53+ cells indeed leads to changes in pre-rRNA processing and, as a consequence, p53 levels – which would be expected as the nucleolar stress response often seen upon ribosome assembly defects.

Additional Northern blot, Western blot, MTT experiments as well as cell cycle analysis under SURF6 knockdown in both HCT116 p53+ and HCT116 p53- cells were performed. These data are presented in new figure 3 and discussed in the revised text:

As it was mentioned before, HeLa cells have defect in p53, meaning that it was not the best choice to study effects of SURF6 on cell cycle. Identification of the potential link between the p53-dependent cell cycle arrest caused by ribosomal stress and cell cycle could be revealed in a pair of HCT116 cell lines which differ only in the expression of the p53 protein (and designated HCT116 p53+ and HCT p53-, respectively). siRNA-mediated knockdown of SURF6 was induced in both cell lines by transient transfection with corresponding control and SURF6-specific siRNA (Fig. 6A). The accumulation of 41S and 18SE precursors occurs in HCT116 p53- cells similar to effects we observed in HeLa cell (Fig. 4G, H). At the same time, 32S, 26S, 21S, 18SE and 12S precursors are downregulated in HCT116 p53+ cell line (Fig. 4H). This result suggests that SURF6 knockdown leads to changes in the level of different pre-rRNA precursors depending on the cell line. As mentioned above, the detailed study of changes in pre-rRNA processing requires RAMP analysis. According to RAMP, SURF6 knockdown in HCT116 p53- cell line causes changes in amounts of pre-rRNAs very similar to the that observed in HeLa cells. The most prominent are the drop in 30S/47S ratio along with the elevated 18S/21S ratio (Fig. 4K). The opposite effect was found in HCT116 p53+ cells subjected to SURF6 knockdown: 30S/47S ratio was higher, while 18SE/41S and 18SE/21S ratios were decreased in comparison to control cells (Fig. 4K). These observations allow assumption that SURF6 knockdown in p53-deficient cells switches the ribosome biogenesis from the pathway 2 to the pathway 1 (Fig. 3 and Fig.4). Opposite to this, the shortage of SURF6 in p53-positive cells facilitates the pre-rRNA processing along the pathway 2 while not causing detectable alterations in overall effectiveness of the process. Arguably, the level of p53 protein in HCT116 p53+ could be affected by SURF6 knockdown in control HCT116 p53+ cells. This possibility was ruled out by evaluation of p53 level in HCT116 p53- and p53+ cell lines following SURF6 knockdown (Fig.6A). One can see that p53 level was not affected by SURF6 knockdown.

Dependence of effects caused by SURF6 on the ribosomal biogenesis and cell cycle on p53 function could be directly proven by comparison of proliferation/viability of p53+ and p53- cells with SURF6 knockdown. We compared the cell cycle distribution of control and SURF6 knockdown cells expressing or lacking p53. In both p53+ and p53- HCT116 SURF6 knockdown caused statistically significant increase in proportion of cells in G0/G1 phase, had no effect on S phase and markedly decreased G2/M phase (Fig. 6B-F). According to MTT-test, proliferation/survival of HCT116 p53+ or p53- cell lines was not negatively influenced by SURF6 knockdown (Fig. 6G). 

Based on these data, we suggest SURF6 affecting cell cycle in HCT116 cells largely in p53-independent manner. According to our guess, the presence of p53 allows cells selecting the optimal ribosome biogenesis pathway to evade cell cycle arrest under the SURF6 shortage conditions.

Other points:

• Figure 1: In agreement with reviewer 1, I also feel that Figure 1 is redundant given that much better data is provided in Figure 2.

We agree with the Reviewer and would like to highlight that the main message of this figure and the chapter is to demonstrate co-localization of SURF6 with ITS1, ITS1, 18S and 28S regions of pre-rRNA transcripts. We added valuable data on localization of SURF6 in knockdown and overexpression experiments and presented better quality quantitative analysis of the RNA intermediates based on Nothern blot and hybridization results. Therefore, we decided to move former figure 1 to supplement section and keep the chapter in the main text to preserve the logic.

• Figure 2: and Supplementary Figure 2: the co-localisation of SURF6 with Fibrillarin and B23 presented in Figure 2 is really informative. Why was a different, less informative method (i.e. ITS1 FISH) used to assess nucleolar structure upon SURF6 OE (Supplementary Figure 2), especially given that the figure does not show a direct comparison of control vs OE cells? It would be beneficial to analyse Fibrillarin and B23 staining upon SURF6 OE vs control cells.

We thank the Reviewer for this comment. We set up additional ICH and confocal microscopy experiments to analyze fibrillarin, B23 and SURF6 localization in the cells with SURF6 OE. The data demonstrate normal structure of the nucleoli upon SURF6 overexpression and expected distribution and staining patterns of fibrillarin and B23 in the nucleoli. These data are presented in new figure 3 and discussed in the revised text:

SURF6 knockdown was induced by transfection of HeLa cells with siRNA specific to human SURF6 mRNA. Changes in SURF6 expression were assessed by immunoblotting of lysates obtained from SURF6 siRNA-transfected and control scramble siRNA-transfected cells according to established protocol [27]. As expected, SURF6 level dramatically decreased in SURF6 siRNA-transfected cells, but not in control cells. We observed at least 90% knockdown efficiency according to analysis of protein level (Fig. 1A, B). In contrast to mouse embryonic fibroblasts, where SURF6 siRNA knockdown caused significant changes in cell morphology and eventual cell death [10], SURF6 knockdown in HeLa had no cytotoxic effect according to MTT test neither disturbed cell morphology (Fig. 1C-R, fig. 5H). For overexpression, pcDNA3.1-hSURF6 was introduced into HeLa cells, and changes in SURF6 level in lysates from transiently transfected cells were examined 48 hours post transfection by Western blotting and immunostaining with anti-SURF6 antibodies. Lysate obtained from HeLa cells transfected in parallel with empty vector was used as negative control. As a result, the amount of SURF6 (expected single 46 kDa band) was greatly increased in lysates from HeLa/pcDNA3.1-SURF6 (Fig. 2A, B). Fluorescence intensity of nucleoli in the HeLa/pcDNA3.1-SURF6 stained with anti-SURF6 was markedly higher than that in control cells (Fig. 2C, G, K, O). According to our estimation, SURF6 level was approximately 5 times higher in HeLa/pcDNA3.1-SURF6 than in control (Fig. 2B). Microscopic images clearly demonstrate that neither overexpression nor knockdown do not affect predominantly nucleolar localization of SURF6 (Fig. 1C-R, fig. 2C-R). It should be noted that despite significant difference in SURF6 level between HeLa with overexpression and knockdown of SURF6, we failed to find detectable changes in size, shape, and pattern of DNA staining in nuclei (Fig. 1C-R, fig. 2C-R). To examine potential changes in fine architecture of nucleolus, we co-stained nucleoli of control HeLa, SURF6 knockdown and overexpression HeLa with anti-SURF6 and anti-fibrillarin (nucleolar fibrillar component marker) or anti-SURF6 and anti-B23 (nucleolar granular component marker). Confocal images demonstrate that fine patterning of fibrillarin and B23 are not affected by both SURF6 knockdown (Fig. 1C-R) and overexpression (Fig.2 C-R). It is interesting that fibrillarin signal is more punctate than that of SURF6 suggesting SURF6 could potentially function outside fibrillar centers [16]. These results suppose that overexpression of human SURF6 in HeLa does not disturb cell morphology in contrast to SURF6 overexpression in mouse embryonic cells [11]. Formally, it is not possible to exclude that there are faint changes in nucleoli of HeLa cells following overexpression or knockdown of SURF6, but their visualization would require high-resolution EM studies.

• Figure 3: the quality of the newly included EtBr-stained gel to show levels of mature 28S and 18S rRNA is poor and not suitable to allow quantitative analysis of the U3 snoRNA, which is then used as a standard to normalise RNA levels. This experiment requires a suitable loading control for normalisation - either an RNA not involved in ribosome assembly, such as the RNA component of RNase P, as mentioned previously, or, at the very least, suitable assessment of mature 28S and 18S rRNA levels by Northern blotting.

We agree with the reviewer that the quality of images that were used as loading and normalization controls (i.e. EtBr stained agarose gel) is poor. Therefore, we performed a new set of Northern blotting experiments and used specific oligo probe to 7SK RNA for proper loading control and normalization. New updated images of Northern hybridization experiments are presented in figure 5.

Page 2, line 59: “In contrast to primates, nucleolus in mice, despite evolutionary closeness of rodents and primates, lacks important structures.”

- This statement requires a reference.

Additional information was added to the main text (lines 61-65) of revised text with corresponding links.

In contrast to primates, nucleolus in mice, despite close evolutionary relationship of rodents and primates, lacks several intrinsic elements. It was found that there is a difference in cleavage at some pre-rRNA sites in human and mouse [12]. In primate’s nucleoli, there are species-specific proteins [13] and lncRNAs [14], which are involved in the organization of the nucleolus structure. Also, some nucleolar proteins, in particular NMP1 (protein partner of SURF6), have been found only in primates [14,15,16]. 

Page 12, line 345: “utilization of E site is prevented by SURF6 depletion”.

- This should say: utilization of E site is promoted by SURF6 depletion”.

Changed to (page 14, lines 404-405):

It is possible that interruption of the LSU maturation causes the disbalance and facilitates the incision at the site E.

Page 12, line 353, 367, 368: “It was hypothesized that cleavages of pre-rRNA at the sites A2 and A3 of yeast ITS1 (correspond to Е and С sites in humans)”.

- Please replace “site C” with “site 2”.

Changed to (page 14, lines 390-392):

It was hypothesized that cleavages of pre-rRNA at the sites A2 and A3 of yeast ITS1 (sites E and 2 in humans) are synchronized with early steps of SSU and LSU assembly.

“Suppression of the SURF6 also inhibits the cleavage at the site C, but not at the site E of ITS1.”

- Please replace “site C” with “site 2”.

Changed to (page 14, lines 403-404):

Suppression of the SURF6 also inhibits the cleavage at the site 2, but not at the site E of ITS1. 

“It has been shown earlier that disturbance of the LSU assembly may affect the ITS1 processing at the site 2c (corresponds to site C in humans) [14].”

- Please replace “site C” with “site 2”.

Changed to (page 14, lines 405-406):

It has been shown earlier that disturbance of the LSU assembly may affect the ITS1 processing at the site 2c in yeast (corresponds to site 2 in humans).

Legend to Figure 5, panel H: “Stars indicate statistically significant differences between control HeLa and cells with knockdown of SURF6 in their viability/proliferation.”

- These are HCT116, not HeLa cells.

The order of figures was changed in the revised version of the manuscript. Figure 5 is now figure 6. Legend to figure 6 was modified accordingly.

(A) SURF6 knockdown efficiency and p53 level in HCT116 p53+ and HCT116 p53- cell lines determined by western blotting. (B-E) Quantitation of DNA content of cells from (A) stained with propidium iodide and analyzed by flow cytometry. Representative histograms show changes in proportions of cells in different stages of the cells cycle in comparison to control cells. (F) Mean percentages of cells in each phase of the cell cycle in four independent experiments. Stars indicate statistically significant differences between control cells HCT116 p53+/SURF6+ or HCT116 p53-/SURF6+, and respective cell lines with siRNA-mediated SURF6 knockdown. (G) MTT assay results obtained in HCT116 p53+/p53- control cells and respective cell lines with SURF6 knockdown and plotted as bar graphs. Mean OD ± SEM values at 570 nm were plotted on the vertical axis. Stars indicate statistically significant differences between control HCT116 and cells with knockdown of SURF6 in their viability/proliferation (p<0,05).

---

## [Editor Report · Decision Letter 2]

3 Feb 2023

PONE-D-22-03434R2Human nucleolar protein SURF6/RRP14 participates in early steps of pre-rRNA processingPLOS ONE

Dear Dr. Moraleva,

Thank you for submitting your manuscript to PLOS ONE. After careful consideration, we feel that it has merit but does not fully meet PLOS ONE’s publication criteria as it currently stands. Therefore, we invite you to submit a revised version of the manuscript that addresses the points raised during the review process. As decided by e-mail communication, I decided to recommend major revision in order you can update the documentation.

We look forward to receiving your revised manuscript.

Kind regards,

Jorge Perez-Fernandez, Ph.D.

Academic Editor

PLOS ONE

Additional Editor Comments (if provided):

I will send to reviewers as soon as I get the updated submission
---

## [Author Response · Author response to Decision Letter 2]

22 Feb 2023

Dear 

Dr. Jorge Perez-Fernandez,

First, my colleagues and I would like to thank you and both reviewers for critical evaluation of our manuscript. We are pleased to receive your kind remarks concerning the quality of data and scientific value of observations described in our study. The issues and concerns raised by the reviewers are reasonable, and it took time and effort to perform some extra experiments and analysis to answer most, if not all, points. The revised version of the manuscript is greatly improved and not only shows the involvement of SURF6 to processing of pre-rRNA, but also provides hints concerning the mechanisms that could be responsible for changes in the cell cycle caused by overexpression or downmodulation of SURF6 in human cells.

Please, find our point-by-point response to reviewers below.

Reviewers’ comments are shown in italic. Our response is in straight font. Changes in the main text of the manuscript are labeled yellow.

Reviewer #1: In the present manuscript, the authors have presented a more comprehensive version of the functional study of Surf6. To test the function of Surf6 in Hela cells, the authors used two complementary approaches, studying the effect of depletion (via a siRNAs strategy) or overexpression of this factor on nucleolus integrity, pre-rRNA processing, and the cell cycle. With the addition of several experiments using new nucleolar markers for nucleolus staining, additional cell lines (HCT-116 p53+ and p53-) and MTT assays, the authors have fully addressed my main comments. Nevertheless, it remains quite intriguing that no proliferation defects are observed following Surf6 depletion in human Hela and HCT-116 cell lines, whereas homologs of this factor are essential in yeast and mouse.

We thank the reviewer for this essential comment. Indeed, human SURF6 has close homologs in yeast and mice. It should be noted that despite evolutionary conservation, the organization of nucleus and smaller structures, especially nucleolus, are quite different between primitive single cell eukaryotes (yeast) and higher vertebrates such as primates and rodents (human and mouse). Thus, at least for yeast and human, we can guess that difference in effects of SURF6 knockdown or overexpression on proliferation could be explained by higher complexity of ribosomal biogenesis in humans and, possibly, development of redundant pathways, which could compensate the loss of one, or several factors involved in the pre-rRNA processing and assembly of subunits. However, this is not the case for human and mouse. Sequence homology between SURF6 and Rpr14 imply that SURF6 may have similar role in biogenesis of mammalian ribosomes. At the same time, there is a lack of information regarding molecular mechanisms that link the process of ribosomal biogenesis with the control of the cell cycle in higher eukaryotes. This makes any parallels between yeast and vertebrates speculative. 

Concerning mouse and humans, we still could not rule out that difference in effects of SURF6 knockdown on cell proliferation may strongly depend on the choice of the cell line. We use immortalized or transformed cells, which usually have defects in control of the cell cycle, which could be related or independent of the p53 function. Another explanation is that rodents and primates have differences in the pathways of rRNA biogenesis. 

We added substantial pieces of text to Discussion section to explain the discrepancy noted by the reviewer.

…Sequence homology between SURF6 and Rpr14 imply that human SURF6 may have similar role in biogenesis of mammalian ribosomes. Function of murine SURF6 was studied in NIH/3T3 fibroblasts [10, 11]. Zatsepina and colleagues used doxycycline inducible system to elevate Surf6 in 3T3 murine embryonic fibroblasts harboring full length Surf6 cDNA or to downmodulate Surf6 in 3T3 cells by transient transfection of genetic construct harboring antisense DNA [10, 11]. Doxycycline-induced Surf6 overexpression in 3T3 cells resulted in increased proliferation rate and altered pre-rRNA processing. Meanwhile, the antisense-mediated knockdown of Surf6 led to growth arrest and higher rate of cell death. These results demonstrated involvement of murine Surf6 in ribosome biogenesis without providing detailed molecular mechanism….

//

…Earlier, it has been reported that SURF6 knockdown in MEF 3T3 leads to elevated cell death in less than 2 days post knockdown induction [10]. In contrast to this phenotype, overexpression of SURF6 in MEF did not cause elevated cell death. 

Apart from this, human nucleolar proteome is represented by at least 4500 proteins with sometimes overlapping or redundant functions (in comparison to yeast S. cerevisiae genome database containing only about 303 nucleolar proteins) suggesting that the ribosomal biogenesis in humans could be more flexible. This fact may help explaining the differences between S. cerevisiae which undergo cell death following RRP14 knockdown and human cells which stay proliferative following SURF6 knockdown….

The authors discuss this observation in the last section of the “Discussion”section, and propose that this could be due to a potential role of Surf6 "in the p53-mediated growth arrest cascade". Nevertheless, the observation that mature 18S and 25S rRNA levels are not affected after Surf6 depletion argues for other hypotheses:

1-For some reason, Surf6 depletion is not strong enough to completely impair pre-rRNA processing, although 90% of the depletion observed by the authors should be sufficient to completely impair ribosome production.

2-In these cells, Surf6 function could also be supported by an additional factor with a redundant function.

3-In these cells, switching from pathway 1 to pathway 2 during pre-rRNA processing does not alter the efficiency of ribosome synthesis.

I think the authors could also insert these concepts into their discussion.

We are grateful to the reviewer for these interesting suggestions and added required concepts to the Discussion section. It should be stressed out that our new data allow making the conclusion that the 3rd scenario could be the likely explanation of the subtle effects on the ribosomal biogenesis efficiency and, hence, on the cell cycle progression. Please see the Discussion section and the text below: 

…Our data suggest that in the absence of functional p53, for instance in human HeLa and HCT116 p53- cells, the shortage of SURF6 forces the usage of the pathway 1 to process pre-rRNA. Both deficit of SURF6 or its overexpression leads to accumulation of 47S/45S and 41S pre-rRNA accompanied by the decrease in 30S and 21S in HeLa cells. Similar, but less profound effects were observed in HCT116 p53- cells with SURF6 siRNA knockdown. (Fig. 4A-K). On the other side, SURF6 knockdown in HCT116 p53+ cells result in opposite effects, and cells switch the ribosome biogenesis to the pathway 2 (Fig. 3, fig. 4C, F, H, K). Our data suggest SURF6 involvement in the cleavage at the site 2 of pre-rRNA 47S/45S that initiates the pathway 2 and cleavage at site 2 in the 41S pre-rRNA in the beginning of the pathway 1. This means also that SURF6 is not required for E site cleavage in 41S and/or 21S pre-rRNAs in both pathways. Apart from this, SURF6 function may be supported with other p53-dependent factor or factors with a redundant function, which allow cells to support pathway 2 of the ribosome biogenesis without notable downregulation of the net effectiveness of this process. This hypothesis is supported by decreased level of 30S pre-rRNA along with the elevation of 18SE pre-rRNA (Fig. 4A-K) in SURF6 knockdown p53- HCT116 cells and the opposite effect on pre-rRNA processing in p53+ HCT116 [19]. 

We conclude that SURF6 may be required for efficient cleavage at the site 2, and in the presence of p53 its function may be supported by other factor(s) which may help cell to choose more effective ribosome biogenesis pathway to adapt to SURF6 shortage. The mechanism of pathways switching remains unknown, but the order of cleavage has been shown to vary depending on species, cell type, physiological and developmental stages. Efficiency of the site E cleavage is not markedly affected by SURF6 knockdown. This interpretation follows from ability of HeLa cells to use bypass way of 18SE pre-rRNA generation and skip intermediates 30S, 26S and 20S (see pathway 2, Fig. 3, Fig. 4). …

…Apart from this, human nucleolar proteome is represented by at least 4500 proteins with sometimes overlapping or redundant functions (in comparison to yeast S. cerevisiae genome database containing only about 303 nucleolar proteins) suggesting that the ribosomal biogenesis in humans could be more flexible. This fact may help explaining the differences between S. cerevisiae which undergo cell death following RRP14 knockdown and human cells which stay proliferative following SURF6 knockdown….

Reviewer #2: The revised manuscript by Moraleva et al. is an improvement on the previous version, which addresses many major points raised, for example with respect to the subnucleolar localisation of SURF6, but some of the other points/newly included experiments require further clarification before publication.

Major concern:

The presented experiments in HeLa and HCT116 cells reveal that the effect of SURF6 KD on cell proliferation is the opposite in human and murine cells. The authors mention that this is likely linked to the lack of p53 in HeLa cells, which normally arrests the cell cycle in response to ribosomal stress. However, while this explanation makes sense for HeLa cells, it does not explain why HCT116 cells are different from murine cells (which express p53 as well), especially given the new data presented in Figure 5 saying that the effect of SURF6 KD on the cell cycle is p53 independent in HCT116 cells.

We would like to thank Reviewer for his time and outstanding expertise. Northern blots and followed RAMP analysis were performed to evaluate the effect of the SURF6 knockdown on the pre-rRNA processing in HCT116 p53+ and HCT116 p53- cells. Following description was added to the main text (lines 322-349 of revised text).

As it was mentioned before, HeLa cells have defect in p53, meaning that it was not the best choice to study effects of SURF6 on cell cycle. Identification of the potential link between the p53-dependent cell cycle arrest caused by ribosomal stress and cell cycle could be revealed in HCT116 cell sublines which differ only in the expression of the p53 protein (and designated HCT116 p53+ and HCT p53-, respectively). siRNA-mediated knockdown of SURF6 was induced in both cell lines by transient transfection with corresponding control and SURF6-specific siRNA (Fig. 6A). The accumulation of 41S and 18SE precursors occurs in HCT116 p53- cells similar to effects we observed in HeLa cell (Fig. 4G, H). At the same time, 32S, 26S, 21S, 18SE and 12S precursors are downregulated in HCT116 p53+ cell line (Fig. 4H). This result suggests that SURF6 knockdown leads to changes in the level of different pre-rRNA precursors depending on the cell line. As mentioned above, the detailed study of changes in pre-rRNA processing requires RAMP analysis. According to RAMP, SURF6 knockdown in HCT116 p53- cell line causes changes in amounts of pre-rRNAs very similar to the that observed in HeLa cells. The most prominent are the drop in 30S/47S ratio along with the elevated 18S/21S ratio (Fig. 4K). The opposite effect was found in HCT116 p53+ cells subjected to SURF6 knockdown: 30S/47S ratio was higher, while 18SE/41S and 18SE/21S ratios were decreased in comparison to control cells (Fig. 4K). These observations allow assumption that SURF6 knockdown in p53-deficient cells switches the ribosome biogenesis from the pathway 2 to the pathway 1 (Fig. 3 and Fig.4). Opposite to this, the shortage of SURF6 in p53-positive cells facilitates the pre-rRNA processing along the pathway 2 while not causing detectable alterations in overall effectiveness of the process. Arguably, the level of p53 protein in HCT116 p53+ could be affected by SURF6 knockdown in control HCT116 p53+ cells. This possibility was ruled out by evaluation of p53 level in HCT116 p53- and p53+ cell lines following SURF6 knockdown (Fig.6A). One can see that p53 level was not affected by SURF6 knockdown.

Dependence of effects caused by SURF6 on the ribosomal biogenesis and cell cycle on p53 function could be directly proven by comparison of proliferation/viability of p53+ and p53- cells with SURF6 knockdown. We compared the cell cycle distribution of control and SURF6 knockdown cells expressing or lacking p53. In both p53+ and p53- HCT116 SURF6 knockdown caused statistically significant increase in proportion of cells in G0/G1 phase, had no effect on S phase and markedly decreased G2/M phase (Fig. 6B-F). According to MTT-test, proliferation/survival of HCT116 p53+ or p53- cell lines was not negatively influenced by SURF6 knockdown (Fig. 6G). 

Based on these data, we suggest SURF6 affecting cell cycle in HCT116 cells largely in p53-independent manner. According to our guess, the presence of p53 allows cells selecting the optimal ribosome biogenesis pathway to evade cell cycle arrest under the SURF6 shortage conditions.

As it stands, the manuscript provides pre-rRNA processing data for HeLa cells, which are not a suitable model for cell cycle analysis. On the other hand, Figure 5 shows interesting cell cycle data for HCT116 p53+ cells – but not enough information to explain how SURF6 levels impact on ribosome assembly/p53 levels/cell cycle/survival etc.

It would therefore be essential to check if SURF6 KD or OE in HCT116 p53+ cells indeed leads to changes in pre-rRNA processing and, as a consequence, p53 levels – which would be expected as the nucleolar stress response often seen upon ribosome assembly defects.

Additional Northern blot, Western blot, MTT experiments as well as cell cycle analysis under SURF6 knockdown in both HCT116 p53+ and HCT116 p53- cells were performed. These data are presented in new figure 3 and discussed in the revised text:

As it was mentioned before, HeLa cells have defect in p53, meaning that it was not the best choice to study effects of SURF6 on cell cycle. Identification of the potential link between the p53-dependent cell cycle arrest caused by ribosomal stress and cell cycle could be revealed in a pair of HCT116 cell lines which differ only in the expression of the p53 protein (and designated HCT116 p53+ and HCT p53-, respectively). siRNA-mediated knockdown of SURF6 was induced in both cell lines by transient transfection with corresponding control and SURF6-specific siRNA (Fig. 6A). The accumulation of 41S and 18SE precursors occurs in HCT116 p53- cells similar to effects we observed in HeLa cell (Fig. 4G, H). At the same time, 32S, 26S, 21S, 18SE and 12S precursors are downregulated in HCT116 p53+ cell line (Fig. 4H). This result suggests that SURF6 knockdown leads to changes in the level of different pre-rRNA precursors depending on the cell line. As mentioned above, the detailed study of changes in pre-rRNA processing requires RAMP analysis. According to RAMP, SURF6 knockdown in HCT116 p53- cell line causes changes in amounts of pre-rRNAs very similar to the that observed in HeLa cells. The most prominent are the drop in 30S/47S ratio along with the elevated 18S/21S ratio (Fig. 4K). The opposite effect was found in HCT116 p53+ cells subjected to SURF6 knockdown: 30S/47S ratio was higher, while 18SE/41S and 18SE/21S ratios were decreased in comparison to control cells (Fig. 4K). These observations allow assumption that SURF6 knockdown in p53-deficient cells switches the ribosome biogenesis from the pathway 2 to the pathway 1 (Fig. 3 and Fig.4). Opposite to this, the shortage of SURF6 in p53-positive cells facilitates the pre-rRNA processing along the pathway 2 while not causing detectable alterations in overall effectiveness of the process. Arguably, the level of p53 protein in HCT116 p53+ could be affected by SURF6 knockdown in control HCT116 p53+ cells. This possibility was ruled out by evaluation of p53 level in HCT116 p53- and p53+ cell lines following SURF6 knockdown (Fig.6A). One can see that p53 level was not affected by SURF6 knockdown.

Dependence of effects caused by SURF6 on the ribosomal biogenesis and cell cycle on p53 function could be directly proven by comparison of proliferation/viability of p53+ and p53- cells with SURF6 knockdown. We compared the cell cycle distribution of control and SURF6 knockdown cells expressing or lacking p53. In both p53+ and p53- HCT116 SURF6 knockdown caused statistically significant increase in proportion of cells in G0/G1 phase, had no effect on S phase and markedly decreased G2/M phase (Fig. 6B-F). According to MTT-test, proliferation/survival of HCT116 p53+ or p53- cell lines was not negatively influenced by SURF6 knockdown (Fig. 6G). 

Based on these data, we suggest SURF6 affecting cell cycle in HCT116 cells largely in p53-independent manner. According to our guess, the presence of p53 allows cells selecting the optimal ribosome biogenesis pathway to evade cell cycle arrest under the SURF6 shortage conditions.

Other points:

• Figure 1: In agreement with reviewer 1, I also feel that Figure 1 is redundant given that much better data is provided in Figure 2.

We agree with the Reviewer and would like to highlight that the main message of this figure and the chapter is to demonstrate co-localization of SURF6 with ITS1, ITS1, 18S and 28S regions of pre-rRNA transcripts. We added valuable data on localization of SURF6 in knockdown and overexpression experiments and presented better quality quantitative analysis of the RNA intermediates based on Nothern blot and hybridization results. Therefore, we decided to move former figure 1 to supplement section and keep the chapter in the main text to preserve the logic.

• Figure 2: and Supplementary Figure 2: the co-localisation of SURF6 with Fibrillarin and B23 presented in Figure 2 is really informative. Why was a different, less informative method (i.e. ITS1 FISH) used to assess nucleolar structure upon SURF6 OE (Supplementary Figure 2), especially given that the figure does not show a direct comparison of control vs OE cells? It would be beneficial to analyse Fibrillarin and B23 staining upon SURF6 OE vs control cells.

We thank the Reviewer for this comment. We set up additional ICH and confocal microscopy experiments to analyze fibrillarin, B23 and SURF6 localization in the cells with SURF6 OE. The data demonstrate normal structure of the nucleoli upon SURF6 overexpression and expected distribution and staining patterns of fibrillarin and B23 in the nucleoli. These data are presented in new figure 3 and discussed in the revised text:

SURF6 knockdown was induced by transfection of HeLa cells with siRNA specific to human SURF6 mRNA. Changes in SURF6 expression were assessed by immunoblotting of lysates obtained from SURF6 siRNA-transfected and control scramble siRNA-transfected cells according to established protocol [27]. As expected, SURF6 level dramatically decreased in SURF6 siRNA-transfected cells, but not in control cells. We observed at least 90% knockdown efficiency according to analysis of protein level (Fig. 1A, B). In contrast to mouse embryonic fibroblasts, where SURF6 siRNA knockdown caused significant changes in cell morphology and eventual cell death [10], SURF6 knockdown in HeLa had no cytotoxic effect according to MTT test neither disturbed cell morphology (Fig. 1C-R, fig. 5H). For overexpression, pcDNA3.1-hSURF6 was introduced into HeLa cells, and changes in SURF6 level in lysates from transiently transfected cells were examined 48 hours post transfection by Western blotting and immunostaining with anti-SURF6 antibodies. Lysate obtained from HeLa cells transfected in parallel with empty vector was used as negative control. As a result, the amount of SURF6 (expected single 46 kDa band) was greatly increased in lysates from HeLa/pcDNA3.1-SURF6 (Fig. 2A, B). Fluorescence intensity of nucleoli in the HeLa/pcDNA3.1-SURF6 stained with anti-SURF6 was markedly higher than that in control cells (Fig. 2C, G, K, O). According to our estimation, SURF6 level was approximately 5 times higher in HeLa/pcDNA3.1-SURF6 than in control (Fig. 2B). Microscopic images clearly demonstrate that neither overexpression nor knockdown do not affect predominantly nucleolar localization of SURF6 (Fig. 1C-R, fig. 2C-R). It should be noted that despite significant difference in SURF6 level between HeLa with overexpression and knockdown of SURF6, we failed to find detectable changes in size, shape, and pattern of DNA staining in nuclei (Fig. 1C-R, fig. 2C-R). To examine potential changes in fine architecture of nucleolus, we co-stained nucleoli of control HeLa, SURF6 knockdown and overexpression HeLa with anti-SURF6 and anti-fibrillarin (nucleolar fibrillar component marker) or anti-SURF6 and anti-B23 (nucleolar granular component marker). Confocal images demonstrate that fine patterning of fibrillarin and B23 are not affected by both SURF6 knockdown (Fig. 1C-R) and overexpression (Fig.2 C-R). It is interesting that fibrillarin signal is more punctate than that of SURF6 suggesting SURF6 could potentially function outside fibrillar centers [16]. These results suppose that overexpression of human SURF6 in HeLa does not disturb cell morphology in contrast to SURF6 overexpression in mouse embryonic cells [11]. Formally, it is not possible to exclude that there are faint changes in nucleoli of HeLa cells following overexpression or knockdown of SURF6, but their visualization would require high-resolution EM studies.

• Figure 3: the quality of the newly included EtBr-stained gel to show levels of mature 28S and 18S rRNA is poor and not suitable to allow quantitative analysis of the U3 snoRNA, which is then used as a standard to normalise RNA levels. This experiment requires a suitable loading control for normalisation - either an RNA not involved in ribosome assembly, such as the RNA component of RNase P, as mentioned previously, or, at the very least, suitable assessment of mature 28S and 18S rRNA levels by Northern blotting.

We agree with the reviewer that the quality of images that were used as loading and normalization controls (i.e. EtBr stained agarose gel) is poor. Therefore, we performed a new set of Northern blotting experiments and used specific oligo probe to 7SK RNA for proper loading control and normalization. New updated images of Northern hybridization experiments are presented in figure 5.

Page 2, line 59: “In contrast to primates, nucleolus in mice, despite evolutionary closeness of rodents and primates, lacks important structures.”

- This statement requires a reference.

Additional information was added to the main text (lines 61-65) of revised text with corresponding links.

In contrast to primates, nucleolus in mice, despite close evolutionary relationship of rodents and primates, lacks several intrinsic elements. It was found that there is a difference in cleavage at some pre-rRNA sites in human and mouse [12]. In primate’s nucleoli, there are species-specific proteins [13] and lncRNAs [14], which are involved in the organization of the nucleolus structure. Also, some nucleolar proteins, in particular NMP1 (protein partner of SURF6), have been found only in primates [14,15,16]. 

Page 12, line 345: “utilization of E site is prevented by SURF6 depletion”.

- This should say: utilization of E site is promoted by SURF6 depletion”.

Changed to (page 14, lines 404-405):

It is possible that interruption of the LSU maturation causes the disbalance and facilitates the incision at the site E.

Page 12, line 353, 367, 368: “It was hypothesized that cleavages of pre-rRNA at the sites A2 and A3 of yeast ITS1 (correspond to Е and С sites in humans)”.

- Please replace “site C” with “site 2”.

Changed to (page 14, lines 390-392):

It was hypothesized that cleavages of pre-rRNA at the sites A2 and A3 of yeast ITS1 (sites E and 2 in humans) are synchronized with early steps of SSU and LSU assembly.

“Suppression of the SURF6 also inhibits the cleavage at the site C, but not at the site E of ITS1.”

- Please replace “site C” with “site 2”.

Changed to (page 14, lines 403-404):

Suppression of the SURF6 also inhibits the cleavage at the site 2, but not at the site E of ITS1. 

“It has been shown earlier that disturbance of the LSU assembly may affect the ITS1 processing at the site 2c (corresponds to site C in humans) [14].”

- Please replace “site C” with “site 2”.

Changed to (page 14, lines 405-406):

It has been shown earlier that disturbance of the LSU assembly may affect the ITS1 processing at the site 2c in yeast (corresponds to site 2 in humans).

Legend to Figure 5, panel H: “Stars indicate statistically significant differences between control HeLa and cells with knockdown of SURF6 in their viability/proliferation.”

- These are HCT116, not HeLa cells.

The order of figures was changed in the revised version of the manuscript. Figure 5 is now figure 6. Legend to figure 6 was modified accordingly.

(A) SURF6 knockdown efficiency and p53 level in HCT116 p53+ and HCT116 p53- cell lines determined by western blotting. (B-E) Quantitation of DNA content of cells from (A) stained with propidium iodide and analyzed by flow cytometry. Representative histograms show changes in proportions of cells in different stages of the cells cycle in comparison to control cells. (F) Mean percentages of cells in each phase of the cell cycle in four independent experiments. Stars indicate statistically significant differences between control cells HCT116 p53+/SURF6+ or HCT116 p53-/SURF6+, and respective cell lines with siRNA-mediated SURF6 knockdown. (G) MTT assay results obtained in HCT116 p53+/p53- control cells and respective cell lines with SURF6 knockdown and plotted as bar graphs. Mean OD ± SEM values at 570 nm were plotted on the vertical axis. Stars indicate statistically significant differences between control HCT116 and cells with knockdown of SURF6 in their viability/proliferation (p<0,05).

---

## [Decision Letter · Decision Letter 3]

7 Mar 2023

PONE-D-22-03434R3Human nucleolar protein SURF6/RRP14 participates in early steps of pre-rRNA processingPLOS ONE

Dear Dr. Moraleva,

Thank you for submitting your manuscript to PLOS ONE. After careful consideration, we feel that it has merit but does not fully meet PLOS ONE’s publication criteria as it currently stands. Therefore, we invite you to submit a revised version of the manuscript that addresses the points raised during the review process.

One of the reviewers raised some valuable comments which need your attention.

We look forward to receiving your revised manuscript.

Kind regards,

Jorge Perez-Fernandez, Ph.D.

Academic Editor

PLOS ONE

Journal Requirements:

Reviewers' comments:

Reviewer's Responses to Questions

**Comments to the Author**

1. If the authors have adequately addressed your comments raised in a previous round of review and you feel that this manuscript is now acceptable for publication, you may indicate that here to bypass the “Comments to the Author” section, enter your conflict of interest statement in the “Confidential to Editor” section, and submit your "Accept" recommendation.

Reviewer #1: All comments have been addressed

Reviewer #2: (No Response)

2. Is the manuscript technically sound, and do the data support the conclusions?

Reviewer #1: Partly

Reviewer #2: Yes

3. Has the statistical analysis been performed appropriately and rigorously? 

Reviewer #1: Yes

Reviewer #2: Yes

4. Have the authors made all data underlying the findings in their manuscript fully available?

Reviewer #1: Yes

Reviewer #2: Yes

5. Is the manuscript presented in an intelligible fashion and written in standard English?

Reviewer #1: Yes

Reviewer #2: Yes

6. Review Comments to the Author

Reviewer #1: In this last version of the manuscript, authors have corrected their manuscript in attempt to answer to our comments. For example, discussion section has been modified to clarify the discrepancies observed after SURF6/Rrp14 depletion in yeast, mice and human cells. Nevertheless, as it stands, it is still rather difficult to interpret some piece of data. Indeed, in both p53+ and p53- HCT_116 cells, SURF6 knock-down promote G1/G0 arrest and a G2/M reduction but do not have any effect on cell death (see MTT assays). Rather, it seems MTT signal is greater after 3 days, indicating a greater proliferation after SURF6 knock-Down in these cells. Although, this work is a good attempt to understand how Surf6 affects ribosome production and/or cell cycle in humans cells. More work will be required to fully decipher the function of this factor.

Reviewer #2: The additional experiments included in the revised manuscript by Moraleva et al. (i.e. the comparison of HCT116 p53+ and p53- cell lines, the use of 7SK as a loading control for Northern Blot quantification and the use of specific marker proteins to assess the sub-nucleolar structure upon SURF6 knockdown/overexpression) have fully addressed my previous concerns.

In agreement with reviewer 1, I also find it quite intriguing that no proliferation defects are observed following SURF6 depletion in human Hela and HCT116 cell lines, as opposed to the situation in yeast and mouse. However, sufficient detail is now provided in the discussion to highlight this discrepancy.

Minor points:

Line 118

….sample (pre-rRNA 34S/36S, 12S/32S etc) …

- According to Figure 3, there is no 34S – please correct.

Line 119

… log2 values of precursors ratios in the control samples (untreated HeLa cells) …

- This should now say (untreated HeLa and HCT116 cells).

Line 128

…. determined by Bredford reaction…

- This should say Bradford.

Line 337

… One can see that p53 level was not affected by SURF6 knockdown.

- This is quite a surprising and important result. The statement should therefore be supported by quantification and statistical analysis of p53 levels in several repeat knockdown experiments.

Line 356

…Sequence homology between SURF6 and Rpr14…

- Rrp14

Line 368

…Sequence homology between human SURF6, murine Surf6 and yeast Rpr14 ...

- Rrp14

Lines 402/403

… It has been shown earlier that disturbance of the LSU assembly may affect the ITS1 processing at the site 2c in yeast (corresponds to site 2 in humans) [19].

- Should this say “processing at the site A3 in yeast (corresponds to site 2 in humans)” ?

7. PLOS authors have the option to publish the peer review history of their article (what does this mean?). If published, this will include your full peer review and any attached files.

Reviewer #1: No

Reviewer #2: No

---

## [Author Response · Author response to Decision Letter 3]

20 Apr 2023

Rebuttal letter for manuscript

PONE-D-22-03434R3

Moraleva et al. Human nucleolar protein SURF6/RRP14 participates in early steps of pre-rRNA processing

PLOS ONE

Dear 

Dr. Jorge Perez-Fernandez,

First, my colleagues and I would like to thank you and both reviewers for critical evaluation of our manuscript. We are pleased to receive these kind remarks concerning the quality of data and scientific value of our study. Concerns raised by the reviewers are reasonable, and some additional experiments and analysis were performed. The revised version of the manuscript contains statistical analysis of the p53 level in HCT116/p53+ cell line comparing cells transfected with scrambled control RNA and siRNA targeting transcript of TP53 gene. Result of this experiment demonstrates no significant difference between control cells and cells with siRNA-induced p53 knockdown. Point-by-point response to minor points is included as well. Please, find our point-by-point response to reviewers below.

Reviewers’ comments are shown in italic. Our responses are in straight font. Changes in the main text of the manuscript are labeled yellow.

Reviewer #1: In this last version of the manuscript, authors have corrected their manuscript in attempt to answer to our comments. For example, discussion section has been modified to clarify the discrepancies observed after SURF6/Rrp14 depletion in yeast, mice and human cells. Nevertheless, as it stands, it is still rather difficult to interpret some piece of data. Indeed, in both p53+ and p53- HCT_116 cells, SURF6 knock-down promote G1/G0 arrest and a G2/M reduction but do not have any effect on cell death (see MTT assays). Rather, it seems MTT signal is greater after 3 days, indicating a greater proliferation after SURF6 knock-Down in these cells. Although, this work is a good attempt to understand how Surf6 affects ribosome production and/or cell cycle in humans cells. More work will be required to fully decipher the function of this factor.

We thank the reviewer for critical comments. In regards to G1/G0 arrest, it is worth noting that observed minor changes in cell numbers likely not sufficient to cause substantial effects on proliferation and/or cell death rates. Apart from this, little is known about possible functions of SURF6 that are not related to ribosome biogenesis. Our data and previous reports show slight increase in proliferation rate due to knockdown of SURF6. These observations are counter intuitive since one would expect the opposite. Nonetheless, we cannot exclude potential secondary/indirect effects caused by changes in SURF6 level on cell proliferation and not linked to ribosome biogenesis. We absolutely agree that our data clearly demonstrate the influence of SURF6 on ribosome biogenesis, but dissection of molecular mechanism underlaying the links between SURF6, p53, ribosome biogenesis and cell cycle regulation goes far beyond the scope of our study. It would require separate study in a near future.

Reviewer #2: The additional experiments included in the revised manuscript by Moraleva et al. (i.e. the comparison of HCT116 p53+ and p53- cell lines, the use of 7SK as a loading control for Northern Blot quantification and the use of specific marker proteins to assess the sub-nucleolar structure upon SURF6 knockdown/overexpression) have fully addressed my previous concerns. In agreement with reviewer 1, I also find it quite intriguing that no proliferation defects are observed following SURF6 depletion in human Hela and HCT116 cell lines, as opposed to the situation in yeast and mouse. However, sufficient detail is now provided in the discussion to highlight this discrepancy.

We thank the reviewer for the careful evaluation of our manuscript. Minor points:

Line 118

….sample (pre-rRNA 34S/36S, 12S/32S etc) …

- According to Figure 3, there is no 34S – please correct.

Changed to pre-rRNA 41S/45-47S, 12S/32S etc. 

Line 119

… log2 values of precursors ratios in the control samples (untreated HeLa cells) …

- This should now say (untreated HeLa and HCT116 cells).

Changed to untreated HeLa and HCT116 cells.

Line 128

…. determined by Bredford reaction…

- This should say Bradford.

Changed to Bradford.

Line 337

… One can see that p53 level was not affected by SURF6 knockdown.

- This is quite a surprising and important result. The statement should therefore be supported by quantification and statistical analysis of p53 levels in several repeat knockdown experiments.

We thank the reviewer for reasonable concern regarding reproducibility of results. Additional experiments were conducted to solve the issue (resulting images of western blot and immunostaining are presented in the raw data file). Statistical analysis was based on three independent siRNA transfection experiments and is shown in figure 6 B. According to the results, there are no statistically significant differences in the p53 level between control cells and cells with SURF6 knockdown. Corresponding changes were made in the legend to figure 6 and experimental section of the manuscript

Line 356

…Sequence homology between SURF6 and Rpr14…

- Rrp14

Changed to Rrp14.

Line 368

…Sequence homology between human SURF6, murine Surf6 and yeast Rpr14 ...

- Rrp14

Changed to Rrp14.

Lines 402/403

… It has been shown earlier that disturbance of the LSU assembly may affect the ITS1 processing at the site 2c in yeast (corresponds to site 2 in humans) [19].

- Should this say “processing at the site A3 in yeast (corresponds to site 2 in humans)” ?

Changed to: “It has been shown earlier that disturbance of the LSU assembly may affect the ITS1 processing at the site A3 in yeast (corresponds to site 2 in humans)”.

---

## [Editor Report · Decision Letter 4]

3 May 2023

Human nucleolar protein SURF6/RRP14 participates in early steps of pre-rRNA processing

PONE-D-22-03434R4

Dear Dr. Moraleva,

We’re pleased to inform you that your manuscript has been judged scientifically suitable for publication and will be formally accepted for publication once it meets all outstanding technical requirements.

Kind regards,

Jorge Perez-Fernandez, Ph.D.

Academic Editor

PLOS ONE

Additional Editor Comments (optional):

As you mentioned in the answer to reviewer #2, please indicate in the corresponding result section or the figure caption that the statistical analysis for p53 expression corresponds to a biological triplicate. You could add these three words when ask to send the definitive manuscript or when you proofread the manuscript.
---

## [Editor Report · Acceptance letter]

6 Jul 2023

PONE-D-22-03434R4 

Human nucleolar protein SURF6/RRP14 participates in early steps of pre-rRNA processing 

Dear Dr. Moraleva:

I'm pleased to inform you that your manuscript has been deemed suitable for publication in PLOS ONE. Congratulations! Your manuscript is now with our production department. 

Kind regards, 

on behalf of

Dr. Jorge Perez-Fernandez 

Academic Editor

PLOS ONE